# DECOY-ENHANCED SALIENCY MAPS

## ABSTRACT

Saliency methods can make deep neural network predictions more interpretable by identifying a set of critical features in an input sample, such as pixels that contribute most strongly to a prediction made by an image classifier. Unfortunately, recent evidence suggests that many saliency methods poorly perform, especially in situations where gradients are saturated, inputs contain adversarial perturbations, or predictions rely upon inter-feature dependence. To address these issues, we propose a framework that improves the robustness of saliency methods by following a two-step procedure. First, we introduce a perturbation mechanism that subtly varies the input sample without changing its intermediate representations. Using this approach, we can gather a corpus of perturbed data samples while ensuring that the perturbed and original input samples follow the same distribution. Second, we compute saliency maps for the perturbed samples and propose a new method to aggregate saliency maps. With this design, we offset the gradient saturation influence upon interpretation. From a theoretical perspective, we show that the aggregated saliency map not only captures inter-feature dependence but, more importantly, is robust against previously described adversarial perturbation methods. Following our theoretical analysis, we present experimental results suggesting that, both qualitatively and quantitatively, our saliency method outperforms existing methods, in a variety of applications.

## 1 INTRODUCTION

Deep neural networks (DNNs) deliver remarkable performance in an increasingly wide range of application domains, but they often do so in an inscrutable fashion, delivering predictions without accompanying explanations. In a practical setting such as automated analysis of pathology images, if a patient sample is classified as malignant, then the physician will want to know which parts of the image contribute to this diagnosis. Thus, in general, a DNN that delivers interpretations alongside its predictions will enhance the credibility and utility of its predictions for end users (Lipton, 2016).

In this paper, we focus on a popular branch of explanation methods, often referred to as *saliency methods*, which aim to find input features (*e.g.,* image pixels or words) that strongly influence the network predictions (Simonyan et al., 2013; Selvaraju et al., 2016; Binder et al., 2016; Shrikumar et al., 2017; Smilkov et al., 2017; Sundararajan et al., 2017; Ancona et al., 2018). Saliency methods typically rely on back-propagation from the network's output back to its input to assign a saliency score to individual features so that higher scores indicate higher importance to the output prediction. Despite attracting increasing attention, saliency methods suffer from several fundamental limitations:

- **Gradient saturation** (Sundararajan et al., 2017; Shrikumar et al., 2017; Smilkov et al., 2017) may lead to the problem that the gradients of important features have small magnitudes, breaking down the implicit assumption that important features, in general, correspond to large gradients. This issue can be triggered when the DNN outputs are flattened in the vicinity of important features.

- **Importance isolation** (Singla et al., 2019) refers to the problem that gradient-based saliency methods evaluate the feature importance in an isolated fashion, implicitly assuming that the other features are fixed.

- **Perturbation sensitivity** (Ghorbani et al., 2017; Kindermans et al., 2017; Levine et al., 2019) refers to the observation that even imperceivable, random perturbations or a simple shift transformation of the input data may lead to a large change in the resulting saliency scores.

In this paper, we tackle these limitations by proposing a decoy-enhanced saliency score. At a high level, our method generates the saliency score of an input by aggregating the saliency scores of multiple perturbed copies of this input. Specifically, given an input sample of interest, our method first generates a population of perturbed samples, referred to as *decoys*, that perfectly mimic the neural network's intermediate representation of the original input. These decoys are used to model the variation of an input sample originating from either sensor noise or adversarial attacks. The decoy construction procedure draws inspiration from the *knockoffs*, proposed recently by Barber & Candès (2015) in the setting of error-controlled feature selection, where the core idea is to generate knockoff features that perfectly mimic the empirical dependence structure among the original features.

In brief, the current paper makes three primary contributions. First, we propose a framework to perturb input samples to produce corresponding decoys that preserve the input distribution, in the sense that the intermediate representations of the original input data and the decoys are indistinguishable. We formulate decoy generation as an optimization problem, applicable to diverse deep neural network architectures. Second, we develop a decoy-enhanced saliency score by aggregating the saliency maps of generated decoys. By design, this score naturally offsets the impact of gradient saturation. From a theoretical perspective, we show how the proposed score can simultaneously reflect the joint effects of other dependent features and achieve robustness to adversarial perturbations. Third, we demonstrate empirically that the decoy-enhanced saliency score outperforms existing saliency methods, both qualitatively and quantitatively, on three real-world applications. We also quantify our method's advantage over existing saliency methods in terms of robustness against various adversarial attacks.

## 2 RELATED WORK

A variety of saliency methods have been proposed in the literature. Some, such as edge detectors and Guided Backpropagation (Springenberg et al., 2014) are independent from the predictive model (Nie et al., 2018; Adebayo et al., 2018). [1] Others are designed only for specific architectures (*i.e.,* Grad-CAM (Selvaraju et al., 2016) for CNNs, DeConvNet for CNNs with ReLU activations (Zeiler & Fergus, 2014)). In this paper, instead of exhaustively evaluating all saliency methods, we apply our method to the three saliency methods that do depend on the predictor (*i.e.,* passing the sanity checks in Adebayo et al. (2018) and Sixt et al. (2020)) and are applicable to diverse DNN architectures:

- The **vanilla gradient** method (Simonyan et al., 2013) simply calculates the gradient of the class score with respect to the input $\mathbf{x}$, which is defined as $E_{grad}(\mathbf{x}; F^c) = \nabla_{\mathbf{x}} F^c(\mathbf{x})$.

- The **SmoothGrad** method (Smilkov et al., 2017) seeks to reduce noise in the saliency map by averaging over explanations of the noisy copies of an input, defined as $E_{sg}(\mathbf{x}; F^c) = \frac{1}{N} \sum_{i=1}^{N} E_{grad}(\mathbf{x} + g_i; F^c)$ with noise vectors $g_i \sim N(0, \sigma^2)$.

- The **integrated gradient** method [2] (Sundararajan et al., 2017) starts from a baseline input $\mathbf{x}^0$ and sums over the gradient with respect to scaled versions of the input ranging from the baseline to the observed input, defined as $E_{ig}(\mathbf{x}; F^c) = (\mathbf{x} - \mathbf{x}^0) \times \int_0^1 \nabla_{\mathbf{x}} F^c(\mathbf{x}^0 + \alpha(\mathbf{x} - \mathbf{x}^0)) d\alpha$.

We do not empirically compare to several other categories of methods. *Counterfactual-based methods* work under the same setup as saliency methods, providing explanations for the predictions of a pre-trained DNN model (Sturmfels et al., 2020). These methods identify the important subregions within an input image by perturbing the subregions (by adding noise, rescaling (Sundararajan et al., 2017), blurring (Fong & Vedaldi, 2017), or inpainting (Chang et al., 2019)) and measuring the resulting changes in the predictions (Ribeiro et al., 2016; Lundberg & Lee, 2017; Chen et al., 2018; Fong & Vedaldi, 2017; Dabkowski & Gal, 2017; Chang et al., 2019; Yousefzadeh & O'Leary, 2019; Goyal et al., 2019). Although these methods do identify meaningful subregions in practice, they exhibit several limitations. First, counterfactual-based methods implicitly assume that regions containing the object most contribute to the prediction (Fan et al., 2017). However, Moosavi-Dezfooli et al. (2017) showed that counterfactual-based methods are also vulnerable to adversarial attacks, which force these methods to output unrelated background rather than the meaningful objects as important subregions.

---

[1] Sixt et al. (2020) shows that LRP (Binder et al., 2016) is independent of the parameters of certain layers.

[2] Ancona et al. (2018) shows that $input \odot gradient$ and DeepLIFT (Shrikumar et al., 2017) are strongly related to the integrated gradient. As such, we only select the integrated gradient.

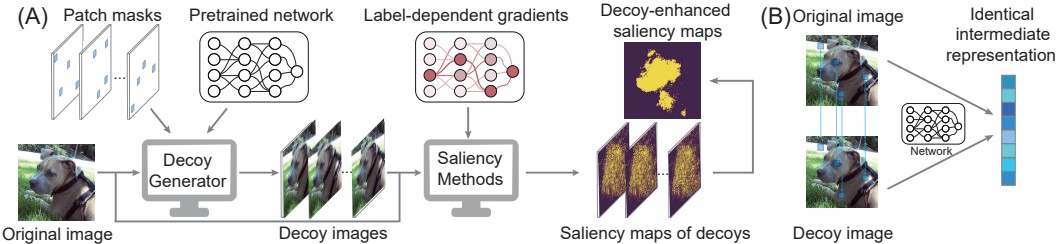

Figure 1: The overview of the proposed method. (A) The workflow of creating decoy-enhanced saliency maps. (B) The operation of swapping image patches between original and decoy images.

Second, the counterfactual images may be potentially far away from the training distribution, causing ill-defined classifier behavior (Burns et al., 2019; Hendrycks & Dietterich, 2019).

In addition to these limitations, counterfactual-based methods and our decoy-based method are fundamentally different in three ways. First, the former seeks the minimum set of features to exclude in order to minimize the prediction score or to include in order to maximize the prediction score (Fong & Vedaldi, 2017), whereas our approach aims to characterize the influence of each feature on the prediction score. Second, counterfactual-based methods explicitly consider the decision boundary by comparing each image to the closest image on the other side of the boundary. In contrast, the proposed method only considers the decision boundary implicitly by calculating the gradient's variants. Third, unlike counterfactual images, which could potentially be out-of-distribution, decoys are plausibly constructed in the sense that their intermediate representations are indistinguishable from the original input data by design. Because of these limitations and differences, we do not compare our method with counterfactual-based methods.

In addition to saliency methods and counterfactual-based methods, several other types of interpretation methods have been proposed that either aim for a different goal or have a different setup. For example, recent research (*e.g.,* Ribeiro et al. (2016); Lundberg & Lee (2017); Chen et al. (2018; 2019b)) designed techniques to explain a black-box model, where the model's internal weights are inaccessible. Koh & Liang (2017) and some follow-up work (Yeh et al., 2018; Koh et al., 2019) tried to find the training points that are most influential for a given test sample. Some other efforts have been made to train a more interpretable DNN classifier (Fan et al., 2017; Zołna et al., 2019; Alvarez-Melis & Jaakkola, 2018; Toneva & Wehbe, 2019), synthesize samples that represent the model predictions (Ghorbani et al., 2019; Chen et al., 2019a)), or identifying noise-tolerant features (Ikeno & Hara, 2018; Schulz et al., 2020). However, due to the task and setup differences, we do not consider these methods in this paper.

## 3 METHODS

### 3.1 PROBLEM SETUP

Consider a multi-label classification task in which a pre-trained neural network model implements a function $F: \mathbb{R}^d \mapsto \mathbb{R}^C$ that maps from the given input $\mathbf{x} \in \mathbb{R}^d$ to $C$ predicted classes. The score for each class $c \in \{1, \cdots, C\}$ is $F^c(\mathbf{x})$, and the predicted class is the one with maximum score, *i.e.,* $\arg\max_{c \in \{1, \cdots, C\}} F^c(\mathbf{x})$. A *saliency method* aims to assign to each feature a *saliency score*, encoded in a saliency map $E(\mathbf{x}; F^c) : \mathbb{R}^d \mapsto \mathbb{R}^d$, in which the features with higher scores represent higher "importance" relative to the final prediction.

Given a pre-trained neural network model $F$ with $L$ layers, an input $\mathbf{x}$, and a saliency method $E$ such that $E(\mathbf{x}; F)$ is a saliency map of the same dimensions as $\mathbf{x}$, the proposed scores can be obtained in two steps: generating decoys and aggregating the saliency maps of the decoys (See Fig. 1(A) that illustrates the workflow of creating decoy-enhanced saliency maps).

## 3.2 DECOY DEFINITION

Say that $F_\ell : \mathbb{R}^d \mapsto \mathbb{R}^{d_\ell}$ is the function instantiated by the given network, which maps from an input $\mathbf{x} \in \mathbb{R}^d$ to its intermediate representation $F_\ell(\mathbf{x}) \in \mathbb{R}^{d_\ell}$ at layer $\ell \in \{1, 2, \cdots, L\}$. A vector $\tilde{\mathbf{x}} \in \mathbb{R}^d$ is said to be a *decoy* of $\mathbf{x} \in \mathbb{R}^d$ at a specified layer $\ell$ if the following swappable condition is satisfied:

$$F_\ell(\mathbf{x}) = F_\ell(\mathbf{x}_{\text{swap}(\tilde{\mathbf{x}}, \mathcal{K})}), \text{ for swappable features } \mathcal{K} \subset \{1, \cdots, d\}. \tag{1}$$

Here, the swap($\tilde{\mathbf{x}}, \mathcal{K}$) operation swaps features between $\mathbf{x}$ and $\tilde{\mathbf{x}}$ based on the elements in $\mathcal{K}$. In this work, $\mathcal{K}$ represents a small meaningful feature set, which represents a small region/segment in an image or a group of words (embeddings) in a sentence. Take an image recognition task for example. Assume $\mathcal{K} = \{10\}$ and $\tilde{\mathbf{x}}$ is a zero matrix, then $\mathbf{x}_{\text{swap}(\tilde{\mathbf{x}}, \mathcal{K})}$ indicates a new image that is identical to $\mathbf{x}$ except that the tenth pixel is set to zero. An illustrative explanation of swap operator is shown in Fig. 1(B). Using the swappable condition, we aim to ensure that the original image $\mathbf{x}$ and its decoy $\tilde{\mathbf{x}}$ are indistinguishable in terms of the intermediate representation at layer $\ell$. Note in particular that the construction of decoys relies solely on the first $\ell$ layers of the neural network $F_1, F_2, \cdots, F_\ell$ and is independent of the succeeding layers $F_{\ell+1}, \cdots, F_L$. As such, $\tilde{\mathbf{x}}$ is conditionally independent of the classification task $F(\mathbf{x})$ given the input $\mathbf{x}$; *i.e.,* $\tilde{\mathbf{x}} \perp\!\!\!\perp F(\mathbf{x})|\mathbf{x}$.

## 3.3 DECOY GENERATION

To identify decoys satisfying the swappable condition, we solve the following optimization problem:

$$\text{maximize}_{\tilde{\mathbf{x}} \in [\mathbf{x}_{\min}, \mathbf{x}_{\max}]^d} \quad \left\| ((\tilde{\mathbf{x}} - \mathbf{x}) \cdot s)^+ \right\|_1,$$
$$\text{s.t.} \begin{cases} \|F_\ell(\tilde{\mathbf{x}}) - F_\ell(\mathbf{x})\|_\infty \leq \epsilon, \\ (\tilde{\mathbf{x}} - \mathbf{x}) \circ (1 - \mathcal{M}) = 0 \end{cases} \tag{2}$$

Here, $(\cdot)^+ = \max(\cdot, 0)$, and the operators $\|\cdot\|_1$ and $\|\cdot\|_\infty$ correspond to the $L_1$ and $L_\infty$ norms, respectively. $\mathcal{M} \in \{0, 1\}^d$ is a specified binary mask. And the value of each feature in the decoy $\tilde{\mathbf{x}}$ is restricted to lie in a legitimate value range *i.e.,* $[\mathbf{x}_{\min}, \mathbf{x}_{\max}]$ (*e.g.,* the pixel value should lie in [0, 255]). We impose the constraint $\|F_\ell(\tilde{\mathbf{x}}) - F_\ell(\mathbf{x})\|_\infty \leq \epsilon$, which ensures that the generated decoy satisfies the swappable condition described in Eqn. 1. It should be noted that we take $\tilde{\mathbf{x}}$ and $\mathbf{x}$ to be indistinguishable except for the swappable features indicated by the mask (*i.e.,* $\mathbf{x}_{swap(\tilde{\mathbf{x}}, \mathcal{K})} = \tilde{\mathbf{x}}$).

As is shown later in Section 3.4, our decoy-enhanced saliency score is defined to capture the empirical range of the decoy saliencies. Here, we first need to estimate the upper/lower ends of the legitimate decoys. To achieve this, in Eqn. 2, we maximize the deviation between $\tilde{\mathbf{x}}$ and $\mathbf{x}$ from both the positive and negative directions, *i.e.,* $s = +1$ and $s = -1$. By using this objective function, for each mask $\mathcal{M}$, we can compute two decoys—one for the positive deviation (*i.e.,* $s = +1$) and the other for the negative one (*i.e.,* $s = -1$).

To solve the optimization function in Eqn. 2, we employ three commonly adopted methods – lagrange multiplier, projected gradient, and change-of-variable (Carlini & Wagner, 2017) – to transform the original objective function into the following form:

$$\text{minimize}_{\hat{\mathbf{x}}} \quad - \left\| \max((\tfrac{1}{2}(\tanh(\hat{\mathbf{x}}) + 1) - \mathbf{x}) \cdot s, 0) \right\|_1 + \lambda \cdot \left\| (|F_\ell(\tfrac{1}{2}(\tanh(\hat{\mathbf{x}}) + 1)) - F_\ell(\mathbf{x})| - \tau)^+ \right\|_2^2, \tag{3}$$

where $\lambda > 0$ is the lagrange multiplier. $\hat{\mathbf{x}}_i = \text{arctanh}(2\tilde{\mathbf{x}}_i - 1)$, for all $i \in \{1, 2, \cdots, d\}$. $\tau > 0$ is introduced to approximate the $L_\infty$ norm in Eqn. 2. After obtaining $\hat{\mathbf{x}}$ by solving Eqn. 3, we compute $\tilde{\mathbf{x}}$ and map it back to the original feature value range $[\mathbf{x}_{\min}, \mathbf{x}_{\max}]$. More details about how to transform Eqn. 2 into Eqn. 3 can be found in Section A6.

## 3.4 DECOY-ENHANCED SALIENCY SCORES

Given an input sample $\mathbf{x}$ and a swappable patch with size $P$, we can obtain $(\sqrt{d} - P + \text{stride})^2$ unique masks by sliding the swappable patch across the input with a certain stride. For computational efficiency, we aggregate $m$ masks into one decoy sample and optimize these masks jointly by solving one decoy sample from Eqn. 3. Then, we can generate $2n$ decoys for that sample. We denote these decoys as $\{\tilde{\mathbf{x}}^1, \tilde{\mathbf{x}}^2, \cdots, \tilde{\mathbf{x}}^{2n}\}$. Here, $n = \lfloor (\sqrt{d} - P + \text{stride})^2 / m \rfloor$. For these decoys,

we can then apply a given saliency method $E$ to yield the corresponding decoy saliency maps $\left\{E(\tilde{\mathbf{x}}^1; F), E(\tilde{\mathbf{x}}^2; F), \cdots, E(\tilde{\mathbf{x}}^{2n}; F)\right\}$. With these decoy saliency maps in hand, for each feature $\mathbf{x}_i$ in $\mathbf{x}$, we can characterize its saliency score variation by using a population of saliency scores $\tilde{E}_i = \left\{E(\tilde{\mathbf{x}}^1; F^c)_i, E(\tilde{\mathbf{x}}^2; F^c)_i, \cdots, E(\tilde{\mathbf{x}}^{2n}; F^c)_i\right\}$. In this work, we define the decoy-enhanced saliency score $Z_i$ for each feature $\mathbf{x}_i$ as

$$Z_i = \max(\tilde{E}_i) - \min(\tilde{E}_i). \tag{4}$$

Here, $Z_i$ is determined by the empirical range of the decoy saliency scores. Ideally, important features will have large values and unimportant ones will have small values. Note that the proposed method is designed specifically for nonlinear models in need of interpretation. As is discussed in Section A7, it cannot output meaningful saliency maps on linear models. It should also be noted that by sliding the swappable patch across the input and ensembling the obtained decoy-enhanced saliency maps, we could capture the saliency of each feature. The motivations of manipulating at a patch level rather than the entire input are capturing the local dependency structure and enabling batch operations for better efficiency.

## 3.5 Theoretical Insights

In this section, we analyze the saliency score method in a theoretical fashion. [3] In particular, we take a convolutional neural network with the ReLU activation function as an example to discuss why the proposed interpretation method can account for inter-feature dependence while also improving explanatory robustness. It should be noted that, while we conduct our theoretical analysis in the setting of CNNs with a specific activation function, the conclusions drawn from the theoretical analysis can easily be extended to other feed-forward neural architectures and other activation functions (e.g., sigmoid and tanh). For analysis of other neural architectures, see Section A9.

Consider a CNN with $L$ hidden blocks, with each layer $\ell$ containing a convolutional layer with a filter of size $\sqrt{s_\ell} \times \sqrt{s_\ell}$ and a max pooling layer with pooling size $\sqrt{s_\ell} \times \sqrt{s_\ell}$. (We set the pooling size the same as the kernel size in each block for simplicity.) The input to this CNN is $\mathbf{x} \in \mathbb{R}^d$, unrolled from a $\sqrt{d} \times \sqrt{d}$ matrix. Similarly, we also unroll each convolutional filter into $\mathbf{g}_\ell \in \mathbb{R}^{s_\ell}$, where $\mathbf{g}_\ell$ is indexed as $(\mathbf{g}_\ell)_j$ for $j \in \mathcal{J}_\ell$. Here, $\mathcal{J}_\ell$ corresponds to the index shift in matrix form from the top-left to bottom-right element. For example, a $3 \times 3$ convolutional filter (i.e., $s_\ell = 9$) is indexed by $\mathcal{J}_\ell = \left\{-\sqrt{d}-1, -\sqrt{d}, -\sqrt{d}+1, -1, 0, 1, \sqrt{d}-1, \sqrt{d}, \sqrt{d}+1\right\}$. The output of the network is the probability vector $\mathbf{p} \in \mathbb{R}^C$ generated by the softmax function, where $C$ is the total number of classes. Such a network can be represented as

$$\begin{aligned}
\mathbf{m}_\ell &= \text{pool}(\text{relu}(\mathbf{g}_\ell * \mathbf{m}_{\ell-1})) \ \text{ for } \ \ell = 1, 2, 3, ..., L, \\
\mathbf{o} &= \mathbf{W}_{L+1}^T \mathbf{m}_L + \mathbf{b}_{L+1}, \\
\mathbf{p} &= \text{softmax}(\mathbf{o}),
\end{aligned} \tag{5}$$

where $\text{relu}(\cdot)$ and $\text{pool}(\cdot)$ indicate the ReLU and pooling operators, $\mathbf{m}_\ell \in \mathbb{R}^{d_\ell}$ is the output of the block $\ell$ ($\mathbf{m}_0 = \mathbf{x}$), and $(\mathbf{g}_\ell * \mathbf{m}_{\ell-1}) \in \mathbb{R}^{d_{\ell-1}}$ represents a convolutional operation on that block. We assume for simplicity that the convolution retains the input shape.

Consider an input $\mathbf{x}$ and its decoy $\tilde{\mathbf{x}}$, generated by swapping features in $\mathcal{K}$. For each feature $i \in \mathcal{K}$, we have the following theorem for the decoy-enhanced saliency score $Z_i$:

**Theorem 1.** In the aforementioned setting, $Z_i$ is bounded by

$$\left| Z_i - \frac{1}{2} \left\| \sum_{k \in \mathcal{K}} (\tilde{\mathbf{x}}_k^+ - \tilde{\mathbf{x}}_k^-)(\mathbf{H}_\mathbf{x})_{k,i} \right\| \right| \leq C_1. \tag{6}$$

Here, $C_1 > 0$ is a bounded constant and $\mathbf{H}_\mathbf{x}$ is the Hessian of $F^c(\mathbf{x})$ on $\mathbf{x}$ where $(\mathbf{H}_\mathbf{x})_{i,k} = \frac{\partial^2 F^c}{\partial \mathbf{x}_i \partial \mathbf{x}_k}$. $\tilde{\mathbf{x}}^+$ and $\tilde{\mathbf{x}}^-$ refer to the decoy that maximizes and minimizes $E(\tilde{\mathbf{x}}; F^c)$, respectively. See Section A7 for the proof. Theorem 1 implies that the proposed saliency score is determined by the second-order Hessian ($(\mathbf{H}_\mathbf{x})_{i,k}$) in the same swappable feature set. The score explicitly models the feature

---

[3] Note that we developed the theoretical properties by using the vanilla gradient as the base saliency method.

dependencies in the swappable feature set via this second-order Hessian, potentially capturing meaningful patterns such as edges, texture, etc.

In addition to enabling representation of inter-feature dependence, Theorem 1 sheds light on the robustness of the proposed saliency score against adversarial attack. To illustrate the robustness improvement of our method, we introduce the following proposition. The proof of this proposition as well as in-depth analysis can be found in Section A8.

**Proposition 1.** Given an input $\mathbf{x}$ and the corresponding adversarial sample $\hat{\mathbf{x}}$, if both $|\mathbf{x}_i - \tilde{\mathbf{x}}_i| \leq C_2\delta_i$ and $\left|\hat{\mathbf{x}}_i - \tilde{\hat{\mathbf{x}}}_i\right| \leq C_2\delta_i$ can be obtain where $C_2 > 0$ is a bounded constant and $\delta_i = |E(\hat{\mathbf{x}}, F)_i - E(\mathbf{x}, F)_i|$, then the following relation can be guaranteed.

$$|(Z_{\hat{\mathbf{x}}})_i - (Z_{\mathbf{x}})_i| \leq |E(\hat{\mathbf{x}}, F)_i - E(\mathbf{x}, F)_i| . \qquad (7)$$

Given an adversarial sample $\hat{\mathbf{x}}$ (*i.e.,* the perturbed $\mathbf{x}$), we say a saliency method is not robust against $\hat{\mathbf{x}}$ if the deviation of the corresponding explanation $\delta_i = |E(\hat{\mathbf{x}}, F)_i - E(\mathbf{x}, F)_i|$ (for all $i \in \{1, 2, \cdots, d\}$) is large. According to the proposition above, we can easily discover that the deviation of our decoy-enhanced saliency score is always no larger than that of other saliency methods when a certain condition is satisfied. This indicates that, when the condition holds, our saliency method can guarantee a stronger resistance to the adversarial perturbation. To ensure the satisfaction of conditions $|\mathbf{x}_i - \tilde{\mathbf{x}}_i| \leq C_2\delta_i$ and $\left|\hat{\mathbf{x}}_i - \tilde{\hat{\mathbf{x}}}_i\right| \leq C_2\delta_i$, we can further introduce the corresponding condition as a constraint to Eqn. 2. In the following section, without further clarification, the saliency scores used in our evaluation are all derived with this constraint imposed.

## 4 Experiments

To evaluate the effectiveness of our proposed method, we perform extensive experiments on deep learning models that target three tasks: image classification, sentiment analysis, and network intrusion detection. The performance of our approach is assessed both qualitatively and quantitatively. The results show that our proposed method identifies intuitively more coherent saliency maps than the state-of-the-art saliency methods alone. The method also achieves quantitatively better alignment to truly important features and demonstrates stronger robustness to adversarial manipulation. The description of the datasets and experimental setup can be found in Section A10.

### 4.1 Saliency benchmark

As mentioned in Section 2, we apply our decoy enhancement method to three saliency methods: vanilla gradient, SmoothGrad, and integrated gradient. Here, we applied the default setup for the integrated gradient (a zero baseline) and SmoothGrad. Section A14 shows that our method can also improve the performance of the variants of the integrated gradient/SmoothGrad and GradCAM. In each case, the decoy-enhanced saliency scores are post-processed in the following way before qualitative and quantitative evaluations. To rule out the bias introduced by the saliency values and ensure a fair comparison, we constructed a binary saliency map by retaining only the top-$K$ features ranked by each method. Specifically, we set the saliency value of the selected features as 1 and the rest features as 0. In this section, we choose the $K$ as the top 20 percent of all features. Note that Section A16 shows that subtly varying $K$ does not influence the experiment conclusions. To demonstrate that all three methods, when enhanced with decoys, still depend on the predictor, we carry out a sanity check on the ImageNet dataset. The results show that our decoy enhanced-saliency methods pass the sanity check (see Section A11 for details).

### 4.2 Performance in various applications

To comprehensively evaluate our proposed approach against the baselines mentioned above, we focus on two criteria. First, we aim to achieve qualitative coherence of the identified saliency map. Intuitively, we prefer a saliency method that highlights features that align closely with the predictions (*e.g.,* highlights the object of interest in an image or the words indicating the sentiment of the sentence). Second, we aim to quantify the correctness of the saliency maps produced by the corresponding method. To do it, we use the fidelity metric (Dabkowski & Gal, 2017), defined as:

$$SF(E(\cdot; F^c), \mathbf{x}) = -\log F^c(E(\mathbf{x}; F^c) \circ \mathbf{x}) \qquad (8)$$

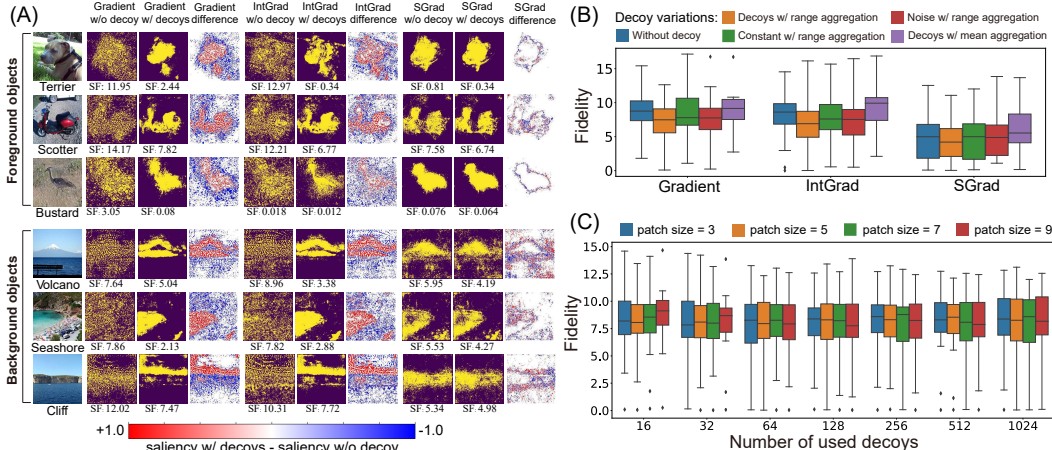

Figure 2: **Performance evaluation on ImageNet.** (A) Visualization of saliency maps on foreground and background objects. (B) Fidelity comparison of original saliency method (*i.e.,* "Without decoys"), our method (*i.e.,* "Decoys w/ range aggregation"), and its alternatives: replacing the decoy generation (Eqn. 2) with constant perturbation (*i.e.,* "Constant w/ range aggregation") or noise perturbation (*i.e.,* "Noise w/ range aggregation"); replacing the decoy aggregation (Eqn. 4) with mean aggregation (*i.e.,* "Decoys w/ mean aggregation") (See Tab. A4 for more statistics about the performance differences between our method and the baselines). (C) Performance with regard to variant patch size and different number of decoys.

where $c$ indicates the predicted class of input $\mathbf{x}$, and $E(\mathbf{x}; F^c)$ is the top-$K$-retained binary saliency map described above. $E(\mathbf{x}; F^c) \circ \mathbf{x}$ performs entry-wise multiplication between $E(\mathbf{x}; F^c)$ and $\mathbf{x}$, encoding the overlap between the object of interest and the concentration of the saliency map. The rationale behind this metric utilization is as follows. By viewing the saliency score of the feature as its contribution to the predicted class, a good saliency method will highlight more important features and thus give rise to higher predicted class scores and lower metric values.

### 4.2.1 PERFORMANCE ON THE IMAGENET DATASET

We applied our decoy-enhanced saliency score to randomly sampled images from the ImageNet dataset (Russakovsky et al., 2015), with a pretrained VGG16 model (Simonyan & Zisserman, 2014). See Section A12 for applicability of our method to diverse CNN architectures such as AlexNet (Krizhevsky et al., 2012) and ResNet (He et al., 2016). The $3 \times 3$ image patches are treated as swappable features in generating decoys.

A side-by-side comparison (Fig. 2(A)) suggests that decoys consistently help to reduce noise and produce more visually coherent saliency maps. For example, the original integrated gradient method highlights the region of dog head in a scattered format, which is also revealed by the difference plot. In contrast, the decoy-enhanced integrated gradient method not only highlights the missing body but also identifies the dog head with more details such as ears, cheek, and nose (See Section A18 for more visualization examples). The visual coherence is also quantitatively supported by the saliency fidelity score.

To further evaluate the necessity of the two steps (*i.e.,* decoy generation and aggregation) in our method, we carried out a control experiment by replacing either step with alternatives. Specifically, as alternatives to the decoy generation, we used an image in which all pixel values are either replaced with a single mean pixel value or contaminated with Gaussian white noise. Regarding the decoy aggregation, we calculated the mean saliency score as the alternative. As shown in Fig. 2(B), our method, which incorporate both steps, reports the best performance. This validates the effectiveness of each of our designs.

Recall that the number of decoys $n$ is decided by the patch size ($P$), stride, and the number of multiple masks in one decoy ($m$). Here, we keep stride as 1 and vary $n$ by selecting different $P$ and

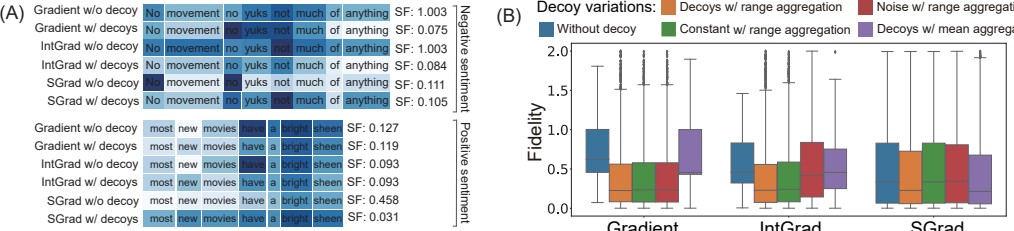

Figure 3: **Evaluation results obtained from the SST dataset.** (A) Visualization of saliency maps in each word, where the normalized saliency values are shown for better distinction. (B) Fidelity comparison of the original saliency method, our method, and its alternatives. Here, the alternative methods represent the practice of replacing the decoy generation (Eqn. 2) with constant perturbation or noise perturbation as well as the practice of replacing the decoy aggregation (Eqn. 4) with mean aggregation (See Tab. A5 for more statistics about the performance differences).

$M$. Fig. 2(C) shows that our method achieves stable fidelity scores across the substantial variations of decoy numbers. The sensitivity test of other hyper-parameters can be found in Section A16.

### 4.2.2 PERFORMANCE ON THE STANFORD SENTIMENT TREEBANK (SST) DATASET

We also applied our decoy-enhanced saliency score to randomly sampled sentences from the Stanford Sentiment Treebank (SST) (Russakovsky et al., 2015). We train a two-layer CNN (Kim, 2014) which takes the pretrained word embeddings as input (Pennington et al., 2014) (see A10 for experimental details). As suggested by Guan et al. (2019), the average saliency value of all dimensions of a word embedding is regarded as the word-level saliency value. The embeddings of the words are treated as swappable features when generating decoys.

As shown in Fig. 3(A), a side-by-side comparison suggests that decoys consistently help to produce semantically more meaningful saliency maps. For example, in a sentence with negative sentiment, keywords associated with negation, such as 'no' and 'not', are more highlighted by decoy-enhanced saliency methods. The semantic coherence is also quantitatively supported by the saliency fidelity (Fig. 3(B)). We also tested the alternatives mentioned above: constant (replacing the decoy generation with the mean embedding of the whole dictionary) and noise perturbation with range aggregation, and decoy with mean aggregation. Fig. 3(B) shows that our method outperforms these alternatives.

### 4.3 ROBUSTNESS TO ADVERSARIAL ATTACKS

Next we investigate the robustness of our method to adversarial manipulations of images. In particular, we focus on three popular adversarial attacks (Ghorbani et al., 2017): (1) the top-$k$ attack, which seeks to decrease the scores of the top $k$ most important features, (2) the target attack, which aims to increase the importance of a pre-specified region in the input image, and (3) the mass-center attack, which aims to spatially change the center of mass of the original saliency map. Here, we specify the bottom-right $4 \times 4$ region of the original image for the target attack and select $k = 5000$ in the top-$k$ attack. We use the sensitivity metric (Alvarez-Melis & Jaakkola, 2018) to quantify the robustness of a saliency method $E$ to adversarial attack, defined as:

$$SS(E(\cdot, F^c), \mathbf{x}, \hat{\mathbf{x}}) = \frac{\|(E(\mathbf{x}, F^c) - E(\hat{\mathbf{x}}, F^c))\|_2}{\|\mathbf{x} - \hat{\mathbf{x}}\|_2} \tag{9}$$

where $\hat{\mathbf{x}}$ is the perturbed image of $\mathbf{x}$. A small sensitivity value means that similar inputs do not lead to substantially different saliency maps.

As shown in Fig. 4(A), a side-by-side comparison suggests that decoys consistently yield low sensitivity scores and help to produce more visually coherent saliency maps, mitigating the impact of various adversarial attacks. More examples can be found in Section A18. The visual coherence and robustness to adversarial attacks are also quantitatively supported by Fig. 4(B)~(D). As is mentioned above, we also did experiments on a MLP trained with a network intrusion dataset and show the results in Section A13. The results are consistent with those on CNNs, which confirm our method's applicability to the widely-used feed-forward networks.

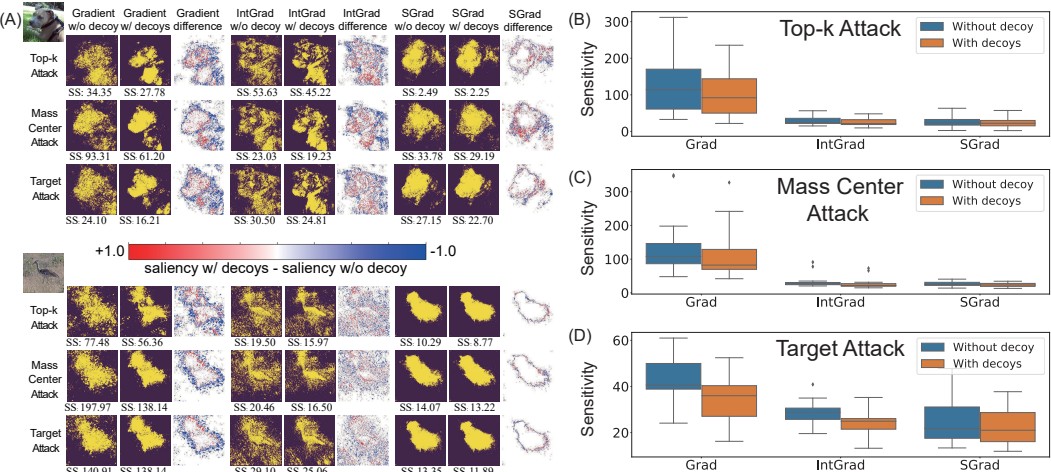

Figure 4: **Robustness to adversarial attacks on images.** (A) Visualization of saliency maps under adversarial attacks. (B)~(D) The decoy-enhanced saliency score is compared to the original saliency score under adversarial attacks, evaluated by sensitivity (See Tab. A6 for more statistics about the performance differences).

## 5    DISCUSSION AND CONCLUSION

In this work, we propose a method for computing, from a given saliency method, decoy-enhanced saliency scores that yield more accurate and robust saliency maps. We formulate the decoy generation as an optimization problem, applicable to diverse deep neural network architecture. We demonstrate the superior performance of our method relative to three standard saliency methods, both qualitatively and quantitatively, even in the presence of various adversarial perturbations to the image. From a theoretical perspective, by deriving a closed-form solution, we show that the proposed score can provably compensate for the limitations of existing saliency methods by reflecting the joint effects from other dependent features and maintaining robustness to adversarial perturbations.

Fig. 2(C) shows our method can achieve a decent performance with only a small number of decoys. Section A15 further shows the runtime of generating one decoy is marginal compared to existing saliency methods. This indicates that our technique can improve the existing saliency methods without introducing too much computational overhead. With the parallel computing enabled by multiple GPUs, our method can be much faster, which further decreases the overhead and escalates our method's practicability (See Section A15 for a detailed runtime analysis). Our method mainly introduce three hyperparameters: swappable feature size $K$, network layer $\ell$, and initial Lagrange multiplier $\lambda$. In Section A16, we show that our method is insensitive to the substantial variation of hyperparameters. We generate decoys by using Eqn. 2. While there are other widely used perturbation methods (*e.g.,* random noise, blurring, and inpainting), they are not suitable for generating decoys. First, Section 4 shows that some general pertrbations (*i.e.,* random noise and constant perturbation) obtain worse fidelity than decoy. Second, without ensuring the swappable condition in Eqn. 1, they cannot provide a theoretical guarantee for robustness improvement. Third, methods like blurring and inpainting are not well-defined for applications beyond computer vision.

This work points to several promising directions for future research. First, $(E(\mathbf{x}; F^c) \circ \mathbf{x})$ may be out-of-distribution and thus fails our fidelity metric. We will investigate more rigorous metrics and use other benchmark datasets (*e.g.,* BAM (Yang & Kim, 2019)) for evaluation. Second, a possible extension is to customize our method to recurrent neural networks and to inputs with categorical/discrete features. Third, recent work (Bansal et al., 2020; Chen et al., 2019c) shows that adversarial training can improve the interpretability of a DNN model. It is worth exploring whether our method could further enhance the quality of saliency maps derived from these adversarially retrained classifiers. A fourth promising direction could be reframing interpretability as hypothesis testing and using decoys to deliver a set of salient features, subject to false discovery rate control at some pre-specified level (Burns et al., 2019; Lu et al., 2018).

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

## A6   IMPLEMENTATION DETAILS

The optimization function proposed to generate decoys is non-differentiable and very difficult to solve; hence, we instead solve an alternate formulation with the help of the following tricks. First, we introduce a Lagrange multiplier $\lambda > 0$ and augment the first constraint in the optimization function as a penalty in the objective function. This will rule out the hyper-parameter $\epsilon$ in Eqn. 2. Second, we use projected gradient descent during the optimization to eliminate the mask constraint (*i.e.,* $(\tilde{\mathbf{x}} - \mathbf{x}) \circ (1 - \mathcal{M}) = 0$). Specifically, after each standard gradient descent step, we enforce $\tilde{\mathbf{x}} = \tilde{\mathbf{x}} \circ \mathcal{M} + \mathbf{x} \circ (1 - \mathcal{M})$. Third, we use the change-of-variable trick (Carlini & Wagner, 2017) to eliminate the feature value constraint (*i.e.,* $\tilde{\mathbf{x}} \in [\mathbf{x}_{\min}, \mathbf{x}_{\max}]^d$). Instead of directly optimizing $\tilde{\mathbf{x}}$, we first normalize it to $[0, 1]$ and introduce $\hat{\mathbf{x}}$ satisfying $\tilde{\mathbf{x}}_i = \frac{1}{2}(\tanh(\hat{\mathbf{x}}_i) + 1)$, for all $i \in \{1, 2, \cdots, d\}$.

Because $\tanh(\hat{\mathbf{x}}_i) \in [-1, 1]$ implies $\tilde{\mathbf{x}}_i \in [0, 1]$, any solution to $\hat{\mathbf{x}}$ is naturally valid. It should be noted that other transformations for this third step are also possible but were not explored in this paper. Putting these ideas together, we minimize the following objective function:

$$\text{minimize}_{\hat{\mathbf{x}}} \quad -\left\|(\frac{1}{2}(\tanh(\hat{\mathbf{x}}) + 1) - \mathbf{x}) \cdot s)^+\right\|_1 + \lambda \cdot \left\|F_\ell(\frac{1}{2}(\tanh(\hat{\mathbf{x}}) + 1)) - F_\ell(\mathbf{x})\right\|_\infty, \quad (10)$$

where $\lambda > 0$ is initialized small and repeatedly doubled until the optimization succeeds. Because the $L_\infty$ norm is not fully differentiable, we adopt the approximation trick introduced by Carlini & Wagner (2017) and solve the following formulation:

$$\text{minimize}_{\hat{\mathbf{x}}} \quad -\left\|\max((\frac{1}{2}(\tanh(\hat{\mathbf{x}}) + 1) - \mathbf{x}) \cdot s, 0)\right\|_1 + \lambda \cdot \left\|(|F_\ell(\frac{1}{2}(\tanh(\hat{\mathbf{x}}) + 1)) - F_\ell(\mathbf{x})| - \tau)^+\right\|_2^2,$$
$$(11)$$

where $\tau > 0$. In this paper, we follow the selection strategy proposed in Carlini & Wagner (2017) and initialize $\tau = 1$. After each iteration, if the second term is zero, then we reduce $\tau$ by a factor of 0.95 and repeat; otherwise, we terminate the optimization. After obtaining $\hat{\mathbf{x}}$, we compute $\tilde{\mathbf{x}}$ and map it back to the original feature value range $[\mathbf{x}_{\min}, \mathbf{x}_{\max}]$. Note that Eqn. 3 can be efficiently solved by any first-order optimization method without introducing too much computational overhead. In practice, the average run time of solving it is 62.3% shorter than the fastest, vanilla gradient method.

## A7    PROOF OF THEOREM 1

Before proving Theorem 1, we first state and prove the following lemma.

**Lemma 1.** *Consider an input $\mathbf{x}$ and its decoy $\tilde{\mathbf{x}}$, generated by replacing the original features with swappable features in $\mathcal{K}$, $|\mathcal{K}| = K$. The partial derivative of $F^c(\tilde{\mathbf{x}})$ w.r.t. to $\tilde{\mathbf{x}}_i$ for $i \in \mathcal{K}$ is*

$$\left|(\nabla_{\tilde{\mathbf{x}}} F^c(\tilde{\mathbf{x}}))_i - \frac{1}{2} \sum_{k \in \mathcal{K}} (\tilde{\mathbf{x}}_k - \mathbf{x}_k)(\mathbf{H}_{\tilde{\mathbf{x}}})_{i,k}\right| \leq C. \quad (12)$$

***Proof.*** The second-order Taylor expansion of the predicted $F^c(\mathbf{x})$ for target class $c$ around $\mathbf{x}$ is as follows:

$$F^c(\mathbf{x}) \approx F^c(\tilde{\mathbf{x}}) + \nabla_{\tilde{\mathbf{x}}} F^c(\tilde{\mathbf{x}})^T \Delta + \frac{1}{2}\Delta^T \mathbf{H}_{\tilde{\mathbf{x}}}\Delta, \quad (13)$$

where $\Delta = \mathbf{x} - \tilde{\mathbf{x}}$. By definition of the decoys in Section 2.2 (*i.e.,* $F^c(\mathbf{x}) = F^c(\tilde{\mathbf{x}})$), the following equation holds:

$$\nabla_{\tilde{\mathbf{x}}} F^c(\tilde{\mathbf{x}})^T \Delta \approx -\frac{1}{2}\Delta^T \mathbf{H}_{\tilde{\mathbf{x}}}\Delta. \quad (14)$$

From the above equation, we can see that, for a linear model, the linearity zeroes out the gradient of the decoys, causing our method to output zero saliency scores for all input features. We clarified in Section 3.4 that our method is mainly defined for non-linear complicated models.

Given a swappable patch of size $K \times 1$ starting from position $i_1$, then $\Delta = [0, ..., \mathbf{x}_{i_1} - \tilde{\mathbf{x}}_{i_1}, ..., \mathbf{x}_{i_K} - \tilde{\mathbf{x}}_{i_K}, 0, ..., 0]$. As such, we have

$$\nabla_{\tilde{\mathbf{x}}} F^c(\tilde{\mathbf{x}})^T \Delta = \sum_{i \in \mathcal{K}} (\nabla_{\tilde{\mathbf{x}}} F^c(\tilde{\mathbf{x}}))_i (\mathbf{x}_i - \tilde{\mathbf{x}}_i),$$
$$\Delta^T \mathbf{H}_{\tilde{\mathbf{x}}}\Delta = \sum_{i \in \mathcal{K}} (\mathbf{x}_i - \tilde{\mathbf{x}}_i) \sum_{k \in \mathcal{K}} (\mathbf{H}_{\tilde{\mathbf{x}}})_{i,k} (\mathbf{x}_k - \tilde{\mathbf{x}}_k). \quad (15)$$

Plugging Eqn. (15) into Eqn. (14), we have

$$\sum_{i \in \mathcal{K}} [(\nabla_{\tilde{\mathbf{x}}} F^c(\tilde{\mathbf{x}}))_i + \frac{1}{2} \sum_{k \in \mathcal{K}} (\mathbf{H}_{\tilde{\mathbf{x}}})_{i,k} (\mathbf{x}_k - \tilde{\mathbf{x}}_k)](\mathbf{x}_i - \tilde{\mathbf{x}}_i) = 0. \quad (16)$$

Then we can derive

$$\left|(\nabla_{\tilde{\mathbf{x}}} F^c(\tilde{\mathbf{x}}))_i + \frac{1}{2} \sum_{k \in \mathcal{K}} (\mathbf{x}_k - \tilde{\mathbf{x}}_k)(\mathbf{H}_{\tilde{\mathbf{x}}})_{i,k}\right| \leq C,$$

$$\left|(\nabla_{\tilde{\mathbf{x}}} F^c(\tilde{\mathbf{x}}))_i - \frac{1}{2} \sum_{k \in \mathcal{K}} (\tilde{\mathbf{x}}_k - \mathbf{x}_k)(\mathbf{H}_{\tilde{\mathbf{x}}})_{i,k}\right| \leq C. \quad (17)$$

First, we can derive $|\tilde{\mathbf{x}}_i - \mathbf{x}_i|$ is bounded by $2\max(\mathbf{x}_{\max}, |\mathbf{x}_{\min}|)$. We also have $|\tilde{\mathbf{x}}_{i+k} - \mathbf{x}_{i+k}|\ 0$ in that we can always find a small perturbation to each feature in $\mathbf{x}$ such that $\|F_\ell(\tilde{\mathbf{x}}) - F_\ell(\mathbf{x})\|_\infty \leq \epsilon$. In addition, both gradient and Hessian are bounded by some Lipschitz constant (Szegedy et al., 2013). [4] As a result, we can always find a constant $C$, such that $C \geq \left| \frac{-\sum_{k_1 \in \mathcal{K} \backslash i}[(\nabla_{\tilde{\mathbf{x}}} F^c(\tilde{\mathbf{x}}))_{k_1} + \frac{1}{2}\sum_{k_2 \in \mathcal{K}}(H_{\tilde{\mathbf{x}}})_{k_1,k_2}(\mathbf{x}_{k_2} - \tilde{\mathbf{x}}_{k_2})](\mathbf{x}_{k_1} - \tilde{\mathbf{x}}_{k_1})}{(\mathbf{x}_i - \tilde{\mathbf{x}}_i)} \right|$. For the case $K = 1$, we have $(\nabla_{\tilde{\mathbf{x}}} F^c(\tilde{\mathbf{x}}))_i = \frac{1}{2}(\mathbf{H}_{\tilde{\mathbf{x}}})_{i,i}(\tilde{\mathbf{x}}_i - \mathbf{x}_i)$.

$\square$

Now we prove Theorem 1 from Section 3.5.

Consider a CNN with $L$ hidden blocks, with each layer $\ell$ containing a convolutional layer with a filter of size $\sqrt{s_\ell} \times \sqrt{s_\ell}$ and a max pooling layer with pooling size $\sqrt{s_\ell} \times \sqrt{s_\ell}$. The input to this CNN is $\mathbf{x} \in \mathbb{R}^d$, unrolled from a $\sqrt{d} \times \sqrt{d}$ matrix. Similarly, we also unroll each convolutional filter into $\mathbf{g}_\ell \in \mathbb{R}^{s_\ell}$, where $\mathbf{g}_\ell$ is indexed as $(\mathbf{g}_\ell)_j$ for $j \in \mathcal{J}_\ell$. Here, $\mathcal{J}_\ell$ corresponds to the index shift in matrix form from the top-left to bottom-right element. The output of the network is the probability vector $\mathbf{p} \in \mathbb{R}^C$ generated by the softmax function, where $C$ is the total number of classes. Such a network can be represented as

$$\mathbf{m}_\ell = \text{pool}(\text{relu}(\mathbf{g}_\ell * \mathbf{m}_{\ell-1})) \text{ for } \ell = 1, 2, 3, ..., L,$$
$$\mathbf{o} = \mathbf{W}_{L+1}^T \mathbf{m}_L + \mathbf{b}_{L+1}, \tag{18}$$
$$\mathbf{p} = \text{softmax}(\mathbf{o}),$$

where $\text{relu}(\cdot)$ and $\text{pool}(\cdot)$ indicate the ReLU and pooling operators, $\mathbf{m}_\ell \in \mathbb{R}^{d_\ell}$ is the output of the block $\ell$ ($\mathbf{m}_0 = \mathbf{x}$), and $(\mathbf{g}_\ell * \mathbf{m}_{\ell-1}) \in \mathbb{R}^{d_{\ell-1}}$ represents a convolutional operation on that block.

Consider an input $\mathbf{x}$ and its decoy $\tilde{\mathbf{x}}$, generated by swapping features in $\mathcal{K}$. For each feature $i \in \mathcal{K}$, we have the following theorem for the decoy-enhanced saliency score $Z_i$:

**Theorem 1.** *In the aforementioned setting, $Z_i$ is bounded by*

$$\left\| Z_i - \frac{1}{2} \left| \sum_{k \in \mathcal{K}} (\tilde{\mathbf{x}}_k^+ - \tilde{\mathbf{x}}_k^-)(\mathbf{H}_{\mathbf{x}})_{k,i} \right| \right\| \leq C_1. \tag{19}$$

*Proof.* The gradient of $\mathbf{p}_c$ with respect to $\mathbf{x}$ can be written as follows, using the denominator layout notation of the derivative of a vector:

$$\nabla_{\mathbf{x}} \mathbf{p}_c = \prod_{\ell=1}^L \frac{\partial \mathbf{m}_\ell}{\partial \mathbf{m}_{\ell-1}} \frac{\partial \mathbf{o}}{\partial \mathbf{m}_L} \frac{\partial \mathbf{p}_c}{\partial \mathbf{o}}, \tag{20}$$

where

$$\frac{\partial \mathbf{o}}{\partial \mathbf{m}_L} = \mathbf{W}_{L+1}, \tag{21}$$

and

$$\begin{cases} \frac{\partial \mathbf{p}_c}{\partial \mathbf{o}_{c'}} = (\mathbf{p}_c - \mathbf{p}_c^2) & \text{if } c' = c, \\ \frac{\partial \mathbf{p}_c}{\partial \mathbf{o}_{c'}} = -\mathbf{p}_c \mathbf{p}_{c'} & \text{otherwise}. \end{cases} \tag{22}$$

Then we can write $\frac{\partial \mathbf{p}_c}{\partial \mathbf{o}}$ as follows:

$$\frac{\partial \mathbf{p}_c}{\partial \mathbf{o}} = \hat{\mathbf{P}}_{\cdot c}, \tag{23}$$

where $\hat{\mathbf{P}}_{\cdot c}$ corresponds to the $c$-th column of $\hat{\mathbf{P}}$ and $\hat{\mathbf{P}} = \text{diag}(\mathbf{p}) - \mathbf{p}\mathbf{p}^T$. We then define $\mathbf{B}_\ell = \frac{\partial \mathbf{m}_\ell}{\partial \mathbf{m}_{\ell-1}}$ as $\mathbf{B}_\ell \in \mathbb{R}^{d_{\ell-1} \times d_\ell}$. In the following, we compute $\mathbf{B}_\ell$.

First, we can have

$$\begin{cases} \frac{\partial (\mathbf{m}_\ell)_j}{\partial (\text{relu}(\mathbf{g}_\ell * \mathbf{m}_{\ell-1}))_n} = 1 & \text{if } \hat{j} - n \in \mathcal{J}_\ell, \text{ and } n = \text{argmax}_{n' \in \hat{j} + \mathcal{J}_\ell} (\mathbf{g}_\ell * \mathbf{m}_{\ell-1})_{n'}, \\ \frac{\partial (\mathbf{m}_\ell)_j}{\partial (\text{relu}(\mathbf{g}_\ell * \mathbf{m}_{\ell-1}))_n} = 0 & \text{otherwise}, \end{cases} \tag{24}$$

---

[4] Following other works that also utilized Lipschitz continuity to analyze DNNs (Szegedy et al., 2013; Ghorbani et al., 2017), we assume that $F_\ell$ is locally continuous around $\mathbf{x}$, for $\ell = 1, 2, ..., L$.

where $\hat{j}$ represents the center of the pooling patch in $\text{relu}(\mathbf{g}_\ell * \mathbf{m}_{\ell-1})$, which results in $(\mathbf{m}_\ell)_j$. Then we can compute

$$\begin{cases} \frac{\partial(\text{relu}(\mathbf{g}_\ell * \mathbf{m}_{\ell-1}))_n}{\partial(\mathbf{m}_{\ell-1})_i} = (a_\ell)_n(\mathbf{g}_\ell)_{n-i} & \text{if } n - i \in \mathcal{J}_\ell\,, \\ \frac{\partial(\text{relu}(\mathbf{g}_\ell * \mathbf{m}_{\ell-1}))_n}{\partial(\mathbf{m}_{\ell-1})_i} = 0 & \text{otherwise}\,, \end{cases} \tag{25}$$

where $(a_\ell)_n = \mathbf{1}\{(\text{relu}(\mathbf{g}_\ell * \mathbf{m}_{\ell-1})_n) \geq 0\}$. If we change the activation function to either sigmoid or tanh, then $(a_\ell)_n$ in Eqn. 25 will be replaced with the derivative of either function. For the sigmoid activation function $\sigma(x)$, the derivative is $\sigma(x)(1 - \sigma(x))$, with a range of $[0, \frac{1}{4}]$. For the tanh activation function $tanh(x)$, the derivative is $1 - tanh(x)^2$, with a range of $[0, 1]$. We conclude that the derivative of both sigmoid and tanh are bounded by a value no larger than 1.

Combining Eqn. 24 with 25, we have

$$\begin{cases} (\mathbf{B}_\ell)_{ij} = \frac{\partial(\mathbf{m}_\ell)_j}{\partial(\mathbf{m}_{\ell-1})_i} = (a_\ell)_n(\mathbf{g}_\ell)_{n-i} & \text{if } n - i \in \mathcal{J}_\ell, \ \hat{j} - n \in \mathcal{J}_\ell, \text{ and } n = \text{argmax}_{n' \in \hat{j} + \mathcal{J}_\ell}(\mathbf{g}_\ell * \mathbf{m}_{\ell-1})_{n'}\,, \\ (\mathbf{B}_\ell)_{ij} = \frac{\partial(\mathbf{m}_\ell)_j}{\partial(\mathbf{m}_{\ell-1})_i} = 0 & \text{otherwise}\,. \end{cases} \tag{26}$$

For simplicity, we rewrite the non-zero condition as $n \in \hat{\mathcal{J}}_\ell$. Plugging $\mathbf{B}_\ell$, $\ell = 1, ..., L$, into Eqn. 20, we can obtain the partial derivative $\nabla_\mathbf{x} \mathbf{p}_c$.

Further, we compute each element in the Hessian matrix $\mathbf{H}_{ij}$ as follows:

$$\mathbf{H}_{ij} = \nabla_{\mathbf{x}_i}(\nabla_{\mathbf{x}_j}\mathbf{p}_c) = \frac{\partial(\prod_{\ell=1}^{L}\mathbf{B}_\ell)_{j\cdot}\mathbf{W}_{L+1}\hat{\mathbf{P}}_{\cdot c}}{\partial\mathbf{x}_i}$$
$$= (\prod_{\ell=1}^{L}\mathbf{B}_\ell)_{j\cdot}\mathbf{W}_{L+1}\frac{\partial\hat{\mathbf{P}}_{\cdot c}}{\partial\mathbf{x}_i} = \left(\sum_{n_L=1}^{d_L}(\prod_{\ell=1}^{L}\mathbf{B}_\ell)_{jn_L}(\mathbf{W}_{L+1})_{n_L\cdot}\right)\frac{\partial\hat{\mathbf{P}}_{\cdot c}}{\partial\mathbf{x}_i}\,, \tag{27}$$

and

$$\frac{\partial\hat{\mathbf{P}}_{c'c}}{\partial\mathbf{x}_i} = \begin{cases} (1 - 2\mathbf{p}_c)\nabla_{\mathbf{x}_i}\mathbf{p}_c & \text{if } c' = c\,, \\ \mathbf{p}_c\nabla_{\mathbf{x}_i}\mathbf{p}_{c'} + \mathbf{p}_{c'}\nabla_{\mathbf{x}_i}\mathbf{p}_c & \text{otherwise}\,. \end{cases} \tag{28}$$

Now we compute $(\prod_{\ell=1}^{L}\mathbf{B}_\ell)_{jn_L}$ as

$$(\prod_{\ell=1}^{L}\mathbf{B}_\ell)_{jn_L} = (B_1)_{j\cdot}\prod_{\ell=2}^{L-1}\mathbf{B}_\ell(\mathbf{B}_L)_{\cdot n_L}\,, \tag{29}$$

where

$$(\mathbf{B}_1)_{j\cdot}\mathbf{B}_2 = [0, ..., C_{n_2}(a_2)_{n_2}\sum_{n_1\in\hat{\mathcal{J}}_1}(a_1)_n\mathbf{g}_{n-1}, ..., 0]\,, \tag{30}$$

and where $C_{n_2} = (g_2)_{n_2-2}\sum_{n_1\in\hat{\mathcal{J}}_1}\mathbf{g}_{n-1}$. Here, we redefine $\hat{\mathcal{J}}_1$ as the set of indices such that $(\mathbf{B}_1)_{jn_1} \neq 0$ for $n_1 \in \hat{\mathcal{J}}_1$. As such, we can compute $(\mathbf{B}_1)_{j\cdot}\prod_{\ell=2}^{L-1}\mathbf{B}_\ell$ as

$$(\mathbf{B}_1)_{j\cdot}\prod_{\ell=2}^{L-1}\mathbf{B}_\ell = [0, .., C_{n_{L-1}}(a_{L-1})_{n_{L-1}}\sum_{\ell=1}^{L-2}\sum_{n_\ell\in\hat{\mathcal{J}}_\ell}(a_\ell)_{n_\ell}, ..., 0]\,. \tag{31}$$

Plugging Eqn. 31 into Eqn. 29, we have

$$(\prod_{\ell=1}^{L}\mathbf{B}_\ell)_{jn_L} = (B_1)_{j\cdot}\prod_{\ell=2}^{L-1}\mathbf{B}_\ell(\mathbf{B}_L)_{\cdot n_L} = (C_L)_{n_L}(a_L)_{n_L}\sum_{\ell=1}^{L-1}\sum_{n_\ell\in\hat{\mathcal{J}}_\ell}(a_\ell)_{n_\ell}\,. \tag{32}$$

Plugging Eqn. 32 into Eqn. 27, we have

$$\mathbf{H}_{ij} = \left(C_j\sum_{\ell=1}^{L}\sum_{n_\ell\in\hat{\mathcal{J}}_\ell}(a_\ell)_{n_\ell}\right)\frac{\partial\hat{\mathbf{P}}_{\cdot c}}{\partial\mathbf{x}_i}\,, \tag{33}$$

where $C_j$ is a linear combination of $\mathbf{g}_1, ..., \mathbf{g}_L, \mathbf{W}_{L+1}$, which is bounded. $\mathbf{H}_{ij}$ equals the multiplication of two components—the summation of neurons activated by $\mathbf{x}$ and a gradient

$\frac{\partial \hat{\mathbf{P}}_{\cdot c}}{\partial \mathbf{x}_i}$. Given the total number of neurons in a CNN is a constant (denoted by $C_T$), we have $0 \leq \left( \sum_{\ell=1}^{L} \sum_{n_\ell \in \hat{\mathcal{J}}_\ell} (a_\ell)_{n_\ell} \right) \leq C_T$. Then, we have $|(\mathbf{H_x})_{ij}| \leq C_T |C_j \frac{\partial \hat{\mathbf{P}}_{\cdot c}}{\partial \mathbf{x}_i}|$. Since the derivatives of both sigmoid and tanh are no larger than 1, this inequality also applies to the network with these two functions as the activation function. Similarly, for the Hessian $(\mathbf{H}_{\tilde{\mathbf{x}}})_{ij}$ of a decoy $\tilde{\mathbf{x}}$, we also have $|(\mathbf{H}_{\tilde{\mathbf{x}}})_{ij}| \leq C_T |C_j \frac{\partial \hat{\mathbf{P}}_{\cdot c}}{\partial \tilde{\mathbf{x}}_i}|$. Given the inequality of $(\mathbf{H}_{\tilde{\mathbf{x}}})_{ij}$ and $(\mathbf{H}_{\tilde{\mathbf{x}}})_{ij}$, we can obtain that $|(\mathbf{H}_{\tilde{\mathbf{x}}})_{ij} - (\mathbf{H_x})_{ij}| \leq 2 C_T \max(|\tilde{C}_j \frac{\partial \hat{\mathbf{P}}_{\cdot c}}{\partial \tilde{\mathbf{x}}_i}|, |C_j \frac{\partial \hat{\mathbf{P}}_{\cdot c}}{\partial \mathbf{x}_i}|)$, where $\frac{\partial \hat{\mathbf{P}}_{\cdot c}}{\partial \mathbf{x}_i}$ is given by Eqn. 28. Recalling that $\mathbf{P}_c$ is within $[0, 1]$, the gradient $\frac{\partial \mathbf{P}_c}{\partial \mathbf{x}_i}$ is bounded by some Lipschitz constant (Szegedy et al., 2013), we can obtain that $\frac{\partial \hat{\mathbf{P}}_{\cdot c}}{\partial \mathbf{x}_i}$ is bounded by some constant. Finally, we can derive that $|(\mathbf{H}_{\tilde{\mathbf{x}}})_{ij} - (\mathbf{H_x})_{ij}| \leq C_C$, where $C_C$ represents the upper bound.[5]

Now, we derive the decoy-enhanced saliency score $Z_i$ for $\mathbf{x}_i$, given a population of saliency scores $\tilde{E}_i = \left\{ E(\tilde{\mathbf{x}}^1; F)_i, E(\tilde{\mathbf{x}}^2; F)_i, \cdots, E(\tilde{\mathbf{x}}^{2n}; F)_i \right\}$. Let $\tilde{\mathbf{x}}^+, \tilde{\mathbf{x}}^- \in \left\{ \tilde{\mathbf{x}}^1, \tilde{\mathbf{x}}^2, \cdots, \tilde{\mathbf{x}}^{2n} \right\}$ denotes the decoy which maximizes and minimize $E(\tilde{\mathbf{x}}; F)_i$, respectively. According to Lemma 1, the partial derivative $\nabla_{\tilde{\mathbf{x}}_i} \mathbf{p}_c$ has the following relationship

$$\left| (\nabla_{\tilde{\mathbf{x}}} F^c(\tilde{\mathbf{x}}))_i - \frac{1}{2} \sum_{k \in \mathcal{K}} (\tilde{\mathbf{x}}_k - \mathbf{x}_k)(\mathbf{H}_{\tilde{\mathbf{x}}})_{i,k} \right| \leq C, \tag{34}$$

Then, we can derive

$$\frac{1}{2} \sum_{k \in \mathcal{K}} (\tilde{\mathbf{x}}_k^+ - \mathbf{x}_k)(\mathbf{H}_{\tilde{\mathbf{x}}^+})_{i,k} - C \leq (\nabla_{\tilde{\mathbf{x}}^+} F^c(\tilde{\mathbf{x}}^+))_i \leq \frac{1}{2} \sum_{k \in \mathcal{K}} (\tilde{\mathbf{x}}_k^+ - \mathbf{x}_k)(\mathbf{H}_{\tilde{\mathbf{x}}^+})_{i,k} + C, \tag{35}$$

$$-\frac{1}{2} \sum_{k \in \mathcal{K}} (\tilde{\mathbf{x}}_k^- - \mathbf{x}_k)(\mathbf{H}_{\tilde{\mathbf{x}}^-})_{i,k} - C \leq -(\nabla_{\tilde{\mathbf{x}}^-} F^c(\tilde{\mathbf{x}}^-))_i \leq -\frac{1}{2} \sum_{k \in \mathcal{K}} (\tilde{\mathbf{x}}_k^- - \mathbf{x}_k)(\mathbf{H}_{\tilde{\mathbf{x}}^-})_{i,k} + C, \tag{36}$$

Then, we have

$$\begin{aligned}
Z_i &= (\nabla_{\tilde{\mathbf{x}}^+} F^c(\tilde{\mathbf{x}}^+))_i - (\nabla_{\tilde{\mathbf{x}}^-} F^c(\tilde{\mathbf{x}}^-))_i \\
&\leq \frac{1}{2} \sum_{k \in \mathcal{K}} (\tilde{\mathbf{x}}_k^+ - \mathbf{x}_k)(\mathbf{H}_{\tilde{\mathbf{x}}^+})_{i,k} - \frac{1}{2} \sum_{k \in \mathcal{K}} (\tilde{\mathbf{x}}_k^- - \mathbf{x}_k)(\mathbf{H}_{\tilde{\mathbf{x}}^-})_{i,k} + 2C \\
&\leq \frac{1}{2} \sum_{k \in \mathcal{K}} (\tilde{\mathbf{x}}_k^+ - \mathbf{x}_k)((\mathbf{H_x})_{i,k} + C_C) - \frac{1}{2} \sum_{k \in \mathcal{K}} (\tilde{\mathbf{x}}_k^- - \mathbf{x}_k)((\mathbf{H}_{\tilde{\mathbf{x}}^-})_{i,k} - C_C) + 2C \\
&\leq \frac{1}{2} \sum_{k \in \mathcal{K}} (\tilde{\mathbf{x}}_k^+ - \tilde{\mathbf{x}}_k^-)(\mathbf{H_x})_{i,k} + \frac{1}{2} C_C \sum_{k \in \mathcal{K}} (\tilde{\mathbf{x}}_k^+ - \tilde{\mathbf{x}}_k^-) + 2C,
\end{aligned} \tag{37}$$

And

$$\begin{aligned}
Z_i &= (\nabla_{\tilde{\mathbf{x}}^+} F^c(\tilde{\mathbf{x}}^+))_i - (\nabla_{\tilde{\mathbf{x}}^-} F^c(\tilde{\mathbf{x}}^-))_i \\
&\geq \frac{1}{2} \sum_{k \in \mathcal{K}} (\tilde{\mathbf{x}}_k^+ - \mathbf{x}_k)(\mathbf{H}_{\tilde{\mathbf{x}}^+})_{i,k} - \frac{1}{2} \sum_{k \in \mathcal{K}} (\tilde{\mathbf{x}}_k^- - \mathbf{x}_k)(\mathbf{H}_{\tilde{\mathbf{x}}^-})_{i,k} - 2C \\
&\geq \frac{1}{2} \sum_{k \in \mathcal{K}} (\tilde{\mathbf{x}}_k^+ - \mathbf{x}_k)((\mathbf{H_x})_{i,k} - C_C) - \frac{1}{2} \sum_{k \in \mathcal{K}} (\tilde{\mathbf{x}}_k^- - \mathbf{x}_k)((\mathbf{H}_{\tilde{\mathbf{x}}^-})_{i,k} + C_C) + 2C \\
&\geq \frac{1}{2} \sum_{k \in \mathcal{K}} (\tilde{\mathbf{x}}_k^+ - \tilde{\mathbf{x}}_k^-)(\mathbf{H_x})_{i,k} - \frac{1}{2} C_C \sum_{k \in \mathcal{K}} (\tilde{\mathbf{x}}_k^+ - \tilde{\mathbf{x}}_k^-) - 2C,
\end{aligned} \tag{38}$$

Combining Eqn. 37 with Eqn. 38, we have

$$\left| Z_i - \frac{1}{2} \left| \sum_{k \in \mathcal{K}} (\tilde{\mathbf{x}}_k^+ - \tilde{\mathbf{x}}_k^-)(\mathbf{H_x})_{k,i} \right| \right| \leq C_1. \tag{39}$$

---

[5]Note that this inequality cannot be directly obtained by the Lipschitz inequality, because the gradient may not be continuous.

Recall that $(\tilde{\mathbf{x}}_k^+ - \tilde{\mathbf{x}}_k^-)$ is bounded by a upper-bound, we can obtain that there exist a constant $C_1$, such that $C_1 \geq \frac{1}{2} C_C \sum_{k \in \mathcal{K}} (\tilde{\mathbf{x}}_k^+ - \tilde{\mathbf{x}}_k^-) + 2C$. Note that this upper bound is data specific, and we leave the exploration on its tightness as a part of future works.

$\square$

## A8  PROOF OF PROPOSITION 1

**Proposition 1.**  *Given an input $\mathbf{x}$ and its corresponding adversarial sample $\hat{\mathbf{x}}$, if both $|\mathbf{x}_i - \tilde{\mathbf{x}}_i| \leq C_2 \delta_i$ and $\left|\hat{\mathbf{x}}_i - \tilde{\hat{\mathbf{x}}}_i\right| \leq C_2 \delta_i$ can obtain where $C_2 > 0$ is a bounded constant and $\delta_i = |E(\hat{\mathbf{x}}, F)_i - E(\mathbf{x}, F)_i|$, then the following relation can be guaranteed.*

$$|(Z_{\hat{\mathbf{x}}})_i - (Z_{\mathbf{x}})_i| \leq |(E(\hat{\mathbf{x}}, F)_i - E(\mathbf{x}, F))_i| . \tag{40}$$

***Proof.*** Recall the goal of the attack against saliency maps is to subtly perturb an input sample such that the added perturbation does not change the output of the classifier (Ghorbani et al., 2017) but force a saliency method to output a less meaningful saliency map (*i.e.,* highlighting features that are irrelevant to the classifier prediction). To achieve this goal, when generating an adversarial sample $\hat{\mathbf{x}}$ from the given input $\mathbf{x}$, an attacker needs to impose the following constraint $\|\hat{\mathbf{x}} - \mathbf{x}\|_\infty \leq \epsilon$. Suppose we have an adversarial sample $\hat{\mathbf{x}}$ satisfies this constraint. Then, we can assume $(\hat{\mathbf{x}} - \mathbf{x})_i = \hat{\epsilon}_i$, where $|\hat{\epsilon}_i| \leq \epsilon$, for $i = 1, 2, ..., d$. In addition, we can compute saliency maps $E(\hat{\mathbf{x}}, F)$ and $E(\mathbf{x}, F)$ for $\hat{\mathbf{x}}$ and $\mathbf{x}$ by using an existing saliency method. [6] Given both saliency maps, we can further compute the difference between $E(\hat{\mathbf{x}}, F)$ and $E(\mathbf{x}, F)$ as

$$(E(\hat{\mathbf{x}}, F) - E(\mathbf{x}, F))_i = \nabla_{\hat{\mathbf{x}}} F^c(\hat{\mathbf{x}}) - \nabla_{\mathbf{x}} F^c(\mathbf{x}) = (\mathbf{H}_\mathbf{x}(\hat{\mathbf{x}} - \mathbf{x}))_i = \sum_{j=1}^{d} (\mathbf{H}_\mathbf{x})_{ij} \hat{\epsilon}_j . \tag{41}$$

Based on the Eqn. 2 in Section 3.3, when generating the decoys $\tilde{\mathbf{x}}$, we ensure the classifier's predictions for those decoys are as same as that of the $\mathbf{x}$. In this work, we achieve this by bounding the difference between the hidden representations of $\tilde{\mathbf{x}}$ and $\mathbf{x}$. As is discussed in Section A7, to preserve the same prediction $c$ for $\tilde{\mathbf{x}}$ and $\mathbf{x}$, one has to ensure $|F^c(\tilde{\mathbf{x}}) - F^c(\mathbf{x})|$ is bounded. This implies the difference between $\tilde{\mathbf{x}}$ and $\mathbf{x}$ is bounded within $\epsilon$. Here, $\epsilon_i$ represents the maximum difference between $\tilde{\mathbf{x}}_i$ and $\mathbf{x}_i$ at the $i^{th}$ dimension. As is mentioned above, the adversarial sample $\hat{\mathbf{x}}$ does not change the classifier's prediction. Therefore, we could imply $\hat{\epsilon}_i \leq \epsilon_i$, for $i = 1, 2, ..., d$.

Now, suppose we obtain a set of decoys for $\mathbf{x}$ and have their corresponding saliency maps, i.e., $\left\{ E(\tilde{\mathbf{x}}^1; F)_i, E(\tilde{\mathbf{x}}^2; F)_i, \cdots, E(\tilde{\mathbf{x}}^{2n}; F)_i) \right\}$. Let $\tilde{\mathbf{x}}^+ \in \left\{ \tilde{\mathbf{x}}^1, \tilde{\mathbf{x}}^2, \cdots, \tilde{\mathbf{x}}^n \right\}$ denote the decoys which maximize $E(\tilde{\mathbf{x}}; F)_i$ and let $\tilde{\mathbf{x}}^-$ denote the decoys which minimize $E(\tilde{\mathbf{x}}; F)_i$. Similarly, we can also have the corresponding decoys $\tilde{\hat{\mathbf{x}}}^-$ and $\tilde{\hat{\mathbf{x}}}^-$ for the adversarial sample $\hat{\mathbf{x}}$ as well as their corresponding saliency maps. With both the decoys and saliency maps for the input sample $\mathbf{x}$ and its adversarial sample $\hat{\mathbf{x}}$, we can compute the difference between $(Z_{\hat{\mathbf{x}}})_i$ and $(Z_{\mathbf{x}})_i$ as

$$
\begin{aligned}
&(Z_{\hat{\mathbf{x}}})_i - (Z_{\mathbf{x}})_i \\
&= \left( E(\tilde{\hat{\mathbf{x}}}^+, F)_i - E(\tilde{\hat{\mathbf{x}}}^-, F)_i \right) - \left( E(\tilde{\mathbf{x}}^+, F)_i - E(\tilde{\mathbf{x}}^-, F)_i \right) \\
&= \left( (\mathbf{H}_\mathbf{x}(\tilde{\hat{\mathbf{x}}}^+ - \mathbf{x}))_i - (\mathbf{H}_\mathbf{x}(\tilde{\hat{\mathbf{x}}}^- - \mathbf{x}))_i \right) - \left( (\mathbf{H}_\mathbf{x}(\tilde{\mathbf{x}}^+ - \mathbf{x}))_i - (\mathbf{H}_\mathbf{x}(\tilde{\mathbf{x}}^- - \mathbf{x}))_i \right) \\
&= \sum_{j=1}^{d} (\mathbf{H}_\mathbf{x})_{ij} \left( (\tilde{\hat{\mathbf{x}}}_j^+ - \tilde{\hat{\mathbf{x}}}_j^-) - (\tilde{\mathbf{x}}_j^+ - \tilde{\mathbf{x}}_j^-) \right) .
\end{aligned}
\tag{42}
$$

---

[6] For simplicity, we use the vanilla gradient method. The conclusion can be generalized to the other saliency methods considered in this paper

To guarantee an improvement in robustness against the adversarial perturbation, we have to ensure that $|(Z_{\hat{\mathbf{x}}})_i - (Z_{\mathbf{x}})_i| - |(E(\hat{\mathbf{x}}, F) - E(\mathbf{x}, F))_i| \leq 0$, for $i = 1, 2..., d$. That is,

$$
\begin{aligned}
&\left| \sum_{j=1}^{d} (\mathbf{H}_{\mathbf{x}})_{ij} \left( (\hat{\tilde{\mathbf{x}}}_j^+ - \hat{\tilde{\mathbf{x}}}_j^-) - (\tilde{\mathbf{x}}_j^+ - \tilde{\mathbf{x}}_j^-) \right) \right| - \left| \sum_{j=1}^{d} (\mathbf{H}_{\mathbf{x}})_{ij} \hat{\boldsymbol{\epsilon}}_j \right| \leq 0, \\
&\left| \sum_{j=1}^{d} (\mathbf{H}_{\mathbf{x}})_{ij} \left( (\hat{\tilde{\mathbf{x}}}_j^+ - \hat{\tilde{\mathbf{x}}}_j^-) - (\tilde{\mathbf{x}}_j^+ - \tilde{\mathbf{x}}_j^-) \right) \right| \leq \left| \sum_{j=1}^{d} (\mathbf{H}_{\mathbf{x}})_{ij} \hat{\boldsymbol{\epsilon}}_j \right|,
\end{aligned}
\tag{43}
$$

As is discussed in Section A7, $|(\mathbf{H}_{\mathbf{x}})_{ij}| \leq C_C$. With this, we can have

$$
\begin{aligned}
&\left| \sum_{j=1}^{d} (\mathbf{H}_{\mathbf{x}})_{ij} \left( (\hat{\tilde{\mathbf{x}}}_j^+ - \hat{\tilde{\mathbf{x}}}_j^-) - (\tilde{\mathbf{x}}_j^+ - \tilde{\mathbf{x}}_j^-) \right) \right| \\
&\leq \sum_{j=1}^{d} |(\mathbf{H}_{\mathbf{x}})_{ij}| \left| (\hat{\tilde{\mathbf{x}}}_j^+ - \hat{\tilde{\mathbf{x}}}_j^-) - (\tilde{\mathbf{x}}_j^+ - \tilde{\mathbf{x}}_j^-) \right| \\
&\leq \sum_{j=1}^{d} C_c \left| (\hat{\tilde{\mathbf{x}}}_j^+ - \hat{\tilde{\mathbf{x}}}_j^-) - (\tilde{\mathbf{x}}_j^+ - \tilde{\mathbf{x}}_j^-) \right|
\end{aligned}
\tag{44}
$$

By plugging Eqn. 44 into Eqn. 43, we conclude that as long as $\left| (\hat{\tilde{\mathbf{x}}}_j^+ - \hat{\tilde{\mathbf{x}}}_j^-) - (\tilde{\mathbf{x}}_j^+ - \tilde{\mathbf{x}}_j^-) \right| \leq \frac{1}{C_c d} \left| \sum_{j=1}^{d} (\mathbf{H}_{\mathbf{x}})_{ij} \hat{\boldsymbol{\epsilon}}_j \right|$, our method could guarantee to improve the robustness against the adversarial perturbations. Let $\delta_i = |E(\hat{\mathbf{x}}, F)_i - E(\mathbf{x}, F)_i|$. If we can ensure that $|\mathbf{x}_i - \tilde{\mathbf{x}}_i| \leq \frac{1}{4C_c d} \delta_i$ and $\left| \hat{\mathbf{x}}_i - \hat{\tilde{\mathbf{x}}}_i \right| \leq \frac{1}{4C_c d} \delta_i$, we can have $\left| \tilde{\mathbf{x}}_j^+ - \tilde{\mathbf{x}}_j^- \right| \leq \frac{1}{2C_c d} \delta_i$ and $\left| \hat{\tilde{\mathbf{x}}}_j^+ - \hat{\tilde{\mathbf{x}}}_j^- \right| \leq \frac{1}{2C_c d} \delta_i$. Thus, the aforementioned condition can be satisfied, i.e., $\left| (\hat{\tilde{\mathbf{x}}}_j^+ - \hat{\tilde{\mathbf{x}}}_j^-) - (\tilde{\mathbf{x}}_j^+ - \tilde{\mathbf{x}}_j^-) \right| \leq \frac{1}{C_c d} \delta_i$. By setting $C_2 = \frac{1}{4C_c d}$, we could obtain the robustness conditions in Proposition 1.

$\square$

## A9  COROLLARY 1

Consider a multilayer perceptron with $L$ fully-connected hidden layers and a decoy swappable size $K \times 1$. The input of this MLP is $\mathbf{x} \in \mathbb{R}^d$. For each hidden layer, we use the ReLU activation function. Similar to the CNN mentioned above, the output of this CNN is $\mathbf{p} \in \mathbb{R}^C$. The network can be represented as:

$$
\begin{aligned}
\mathbf{m}_\ell &= \text{relu}(\mathbf{W}_\ell^T \mathbf{m}_{\ell-1} + \mathbf{b}_\ell), \quad \text{For } \ell = 1, 3, ..., L, \\
\mathbf{o} &= \mathbf{W}_{L+1}^T \mathbf{m}_L + \mathbf{b}_{L+1}, \\
\mathbf{p} &= \text{softmax}(\mathbf{o}).
\end{aligned}
\tag{45}
$$

where $\mathbf{W}_\ell \in \mathbb{R}^{d_{\ell-1} \times d_\ell}$, for $\ell \in \{1, \cdots, L+1\}$ represents the weights of the neural network, and $\mathbf{b}_\ell \in \mathbb{R}^{d_\ell}$ represents the biases, where $d_0 = d$ and $d_{L+1} = C$. $\mathbf{m}_\ell \in \mathbb{R}^{d_\ell}$ is the output of each hidden layer, with $\mathbf{m}_0 = \mathbf{x}$ and $\mathbf{o} \in \mathbb{R}^C$ is the logits. The entry-wise softmax operator for target class $c$ is defined as $\mathbf{p}_c = \frac{e^{\mathbf{o}_c}}{\sum_{c'=1}^{C} e^{\mathbf{o}_{c'}}}$, for $c \in \{1, 2, \cdots, C\}$.

**Corollary 1.** *For the above MLP, $Z_i$ is also bounded by:*

$$
Z_i \leq \left| \frac{1}{2} \sum_{k \in \mathcal{K}} (\tilde{\mathbf{x}}_{i+k} - \mathbf{x}_{i+k})(\mathbf{H}_{\mathbf{x}})_{i+k,i} \right| + C_2.
\tag{46}
$$

***Proof.*** Based on the proof of Theorem 1, the gradient of $\mathbf{p}_c$ with respect to $\mathbf{x}$ can be written as follows

$$
\nabla_{\mathbf{x}} \mathbf{p}_c = \prod_{l=1}^{L} \mathbf{B}_\ell \mathbf{W}_{L+1} \hat{\mathbf{P}}_{\cdot c}.
\tag{47}
$$

where $\mathbf{B}_\ell = \frac{\partial \mathbf{m}_\ell}{\partial \mathbf{m}_{\ell-1}}$, $\mathbf{B}_\ell \in \mathbb{R}^{d_{\ell-1} \times d_\ell}$. $\hat{\mathbf{P}}_{\cdot c}$ is also defined as $\hat{P} = \text{diag}(\mathbf{p}) - \mathbf{pp}^T$. In the following, we compute $\mathbf{B}_l$. First, we can compute $(\mathbf{B}_1)_{ij}$, in which

$$(\mathbf{B}_1)_{ij} = \frac{\partial (\mathbf{m}_1)_j}{\partial \mathbf{x}_i} = \frac{\partial (\mathbf{W}_1^T \mathbf{x} + \mathbf{b}_1)_j}{\partial \mathbf{x}_i} \frac{\partial (\mathbf{m}_1)_j}{\partial (\mathbf{W}_1^T \mathbf{x} + \mathbf{b}_1)_j} = (W_1)_{ij}(a_1)_j, \tag{48}$$

where $(a_1)_j = \mathbf{1}\{(\mathbf{W}_1^T \mathbf{x} + \mathbf{b}_1)_j \geq 0\}$. Similar, we can also compute $(\mathbf{B}_\ell)_{ij}$, for $\ell = 2, 3, ..., L$

$$(\mathbf{B}_\ell)_{ij} = (W_\ell)_{ij}(a_\ell)_j, \tag{49}$$

where $(a_\ell)_j = \mathbf{1}\{(\mathbf{W}_\ell^T \mathbf{x} + \mathbf{b}_\ell)_j \geq 0\}$.

Then, we compute the each element in the Hessian matrix $\mathbf{H}_{ij}$. Specifically, based on Eqn. 27, we have

$$\mathbf{H}_{ij} = \left( \sum_{n_L=1}^{d_L} (\prod_{\ell=1}^{L} \mathbf{B}_\ell)_{jn_L} (\mathbf{W}_{L+1})_{n_L \cdot} \right) \frac{\partial \hat{\mathbf{P}}_{\cdot c}}{\partial \mathbf{x}_i}, \tag{50}$$

where $\frac{\partial \hat{\mathbf{P}}_{\cdot c}}{\partial \mathbf{x}_i}$ is the same with Eqn. 28.

Now, we compute $(\prod_{\ell=1}^{L} \mathbf{B}_l)_{jn_L}$ as

$$(\prod_{\ell=1}^{L} \mathbf{B}_\ell)_{jn_L} = (B_1)_{j \cdot} \prod_{\ell=2}^{L-1} \mathbf{B}_\ell (\mathbf{B}_L)_{\cdot n_L}, \tag{51}$$

where $(\mathbf{B}_1)_{j \cdot} = [(\mathbf{W}_1)_{j1}(a_1)_1, (\mathbf{W}_1)_{j2}(a_1)_2, ..., (\mathbf{W}_1)_{jd_1}(a_1)_{d_1}]$ and

$$(\mathbf{B}_1)_{j \cdot} \mathbf{B}_2 = [(a_2)_1 \sum_{n_1=1}^{d_1} (C_2)_{1n_1}(a_1)_{n_1}, ..., (a_2)_{d_2} \sum_{n_1=1}^{d_1} (C_2)_{d_2, n_1}(a_1)_{n_1}], \tag{52}$$

where $(C_2)_{n_2, n_1} = (\mathbf{W}_2)_{n_1, n_2}(\mathbf{W}_1)_{j, n_1}$. For simplicity, we can rewrite $\sum_{n_1=1}^{d_1} (C_2)_{n_2, n_1}(a_1)_{n_1} = (C_2)_{n_2} \sum_{n_1=1}^{d_1} (a_1)_{n_1}$. Then, we have

$$(\mathbf{B}_1)_{j \cdot} \mathbf{B}_2 = [(C_2)_1(a_2)_1 \sum_{n_1=1}^{d_1} (a_1)_{n_1}, ..., (C_2)_{d_2}(a_2)_{d_2} \sum_{n_1=1}^{d_1} (a_1)_{n_1}]. \tag{53}$$

As such, we can compute $(\mathbf{B}_1)_{j \cdot} \prod_{\ell=2}^{L-1} \mathbf{B}_\ell$ as

$$(\mathbf{B}_1)_{j \cdot} \prod_{\ell=2}^{L-1} \mathbf{B}_\ell = [(C_{L-1})_1(a_{L-1})_1 \sum_{\ell=1}^{L-2} \sum_{n_\ell=1}^{d_\ell} (a_\ell)_{n_\ell}, ..., (C_{L-1})_{d_{L-1}}(a_{L-1})_{d_{L-1}} \sum_{\ell=1}^{L-2} \sum_{n_\ell=1}^{d_\ell} (a_\ell)_{n_\ell}]. \tag{54}$$

Plugging Eqn. 54 into Eqn. 23, we have

$$(\prod_{\ell=1}^{L} \mathbf{B}_\ell)_{jn_L} = (B_1)_{j \cdot} \prod_{\ell=2}^{L-1} \mathbf{B}_\ell (\mathbf{B}_L)_{\cdot n_L} = (C_L)_{n_L}(a_L)_{n_L} \sum_{\ell=1}^{L-1} \sum_{n_\ell=1}^{d_\ell} (a_\ell)_{n_\ell}. \tag{55}$$

Finally, we can obtain that

$$\begin{aligned}
\mathbf{H}_{ij} &= \left( \sum_{n_L=1}^{d_L} (C_L)_{n_L}(a_L)_{n_L} \sum_{\ell=1}^{L-1} \sum_{n_\ell=1}^{d_\ell} (a_\ell)_{n_\ell} (\mathbf{W}_{L+1})_{n_L \cdot} \right) \frac{\partial \hat{\mathbf{P}}_{\cdot c}}{\partial \mathbf{x}_i} \\
&= \left( C_j \sum_{\ell=1}^{L} \sum_{n_\ell=1}^{d_\ell} (a_\ell)_{n_\ell} \right) \frac{\partial \hat{\mathbf{P}}_{\cdot c}}{\partial \mathbf{x}_i},
\end{aligned} \tag{56}$$

where $C_j$ is a linear combination of the elements in $(\mathbf{W}_1)_{j \cdot}, \mathbf{W}_2, ..., \mathbf{W}_{L+1}$.

Note that the Hessian derived from the MLP has a similar form with the Hessian derived from the CNN in Eqn. 33, i.e., the summation of neurons activated by $\mathbf{x}$ multiplying the gradient. Here, the summation of neurons activated by $\mathbf{x}$ is again bounded by the total number of neurons in the

network. The gradient $\frac{\partial \hat{\mathbf{P}}_{\cdot c}}{\partial \mathbf{x}_i}$ is bounded by a Lipschitz constant. Similarly, we also have the following inequality for $(\mathbf{H}_{\tilde{\mathbf{x}}})_{ij}$ and $(\mathbf{H}_{\mathbf{x}})_{ij}$, i.e., $|(\mathbf{H}_{\tilde{\mathbf{x}}})_{ij} - (\mathbf{H}_{\mathbf{x}})_{ij}| \leq C_M$.

Similar to Theorem 1, let $\tilde{\mathbf{x}}^+, \tilde{\mathbf{x}}^- \in \{\tilde{\mathbf{x}}^1, \tilde{\mathbf{x}}^2, \cdots, \tilde{\mathbf{x}}^{2n}\}$ denotes the decoy which maximizes and minimize $E(\tilde{\mathbf{x}}; F)_i$, respectively. Based on Eqn. 34 to Eqn. 39, we have

$$\left| Z_i - \frac{1}{2} \left| \sum_{k \in \mathcal{K}} (\tilde{\mathbf{x}}_k^+ - \tilde{\mathbf{x}}_k^-)(\mathbf{H}_{\mathbf{x}})_{k,i} \right| \right| \leq C_2 . \tag{57}$$

$C_2 \geq \frac{1}{2}C_M \sum_{k \in \mathcal{K}}(\tilde{\mathbf{x}}_k^+ - \tilde{\mathbf{x}}_k^-) + 2C$. Slightly different for CNN, MLP sometimes is used to process the input that does not have a strong local dependency. In this case, we can set the swappable path size $K = 1$. Then, Eqn. 57 can reformulated as $\left| Z_i - \frac{1}{2} \left| (\tilde{\mathbf{x}}_i^+ - \tilde{\mathbf{x}}_i^-)(\mathbf{H}_{\mathbf{x}})_{i,i} \right| \right| \leq C_2$. As we can observe from this equation, our proposed saliency score is still able to compensate for the gradient saturation problem. □

Table A1: The hyper-parameter choices of the proposed method on different target models.

| | $\ell$ | $\lambda$ | patch_size ($P$) | stride | $\tau$ |
|---|---|---|---|---|---|
| ImageNet AlexNet | 6 | 10000 | 3 | 1 | 1 |
| ImageNet VGG16 | 3 | 10000 | 3 | 1 | 1 |
| ImageNet ResNet | 2 | 10000 | 3 | 1 | 1 |
| SST CNN | 2 | 10000 | 1 | 1 | 1 |
| IDS MLP | 2 | 10000 | 1 | 1 | 1 |

## A10 DATASETS AND EXPERIMENT SETUP

In this section, we introduce the datasets used in our experiments and the neural network trained on each dataset, followed by our choices of hyper-parameters when explaining each model.

**ImageNet.** We randomly select a subset of samples from the ImageNet validation set, which can be downloaded from the following link: http://www.image-net.org/. We adopt the most widely used preprocessing method for the selected images. Specifically, for each image, we resized it to $227 \times 227$, converted it to BGR format, and subtract the mean value of each channel [103.939, 116.779, 123.68] from the image. Rather than training our own networks, we downloaded a pretrained VGG16 model, AlexNet model, and ResNet_v1_50 model from the following link: https://github.com/tensorflow/models/tree/master/research/slim and http://www.cs.toronto.edu/~guerzhoy/tf_alexnet/. We applied our proposed method to explain the predictions of these networks on the selected samples.

**SST.** We downloaded the Stanford Sentiment Treebank (SST1) from the following link: https://github.com/harvardnlp/sent-conv-torch/tree/master/data. The data is spited into a training set of $76,961$ samples and a testing set of $1,821$ samples. We used a pretrained glove embedding to represent each word in the sentences (sample). The embedding of each word is a vector of $100$ dimensions. The pretrained embedding matrix can be downloaded from the following link: http://nlp.stanford.edu/data/wordvecs/glove.6B.zip. We trained a two-layer CNN with the embeddings as inputs. The model achieves about $80\%$ accuracy on the testing set. The preprocessed testing data and the pretrained model can be downloaded from the following link: https://tinyurl.com/y9noqj6l. We run our explanation method on the pretrained model with the testing samples.

**Network intrusion detection (IDS).** We use a subset of CSE-CIC-IDS2018 dataset (Sharafaldin et al., 2018; for Cybersecurity, 2018), a network intrusion dataset contains the benign network traffic traces and malicious traces generated by three types of attacks: Denial of Service (DoS)-Hulk, SSH-BruteForce, and Infiltration. The training set contains $88,661$ samples and the testing set has $22,165$ samples. Each sample is represented as a vector of $83$ dimensions, where each feature represents the statistics of network traffic flows (*e.g.,* Number of packets, Number of bytes, Length of packets, etc). The features are normalized within [0, 1] by using the scikit-learn MinMaxScaler function. We trained a two-layer MLP to classify whether an input is a benign traffic or an attack (intrusion).

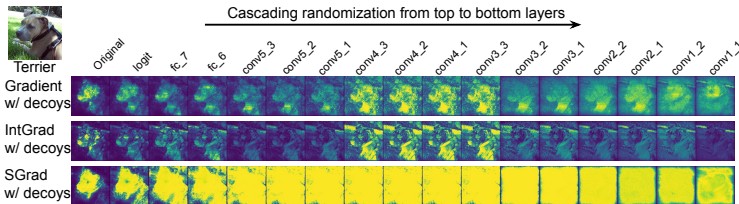

Figure A5: Cascading randomization on VGG16 network. The figure shows the original saliency map (first column) for the terrier. Progression from left to right corresponds to complete randomization of the pretrained VGG16 network weights from the top layer to the bottom layer. Note that, here, we followed the visualization method in Adebayo et al. (2018) to show the saliency maps, i.e., 0-1 normalization. The row labels share the same meanings as the column labels in Fig. 2.

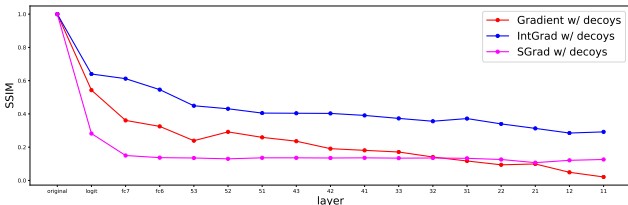

Figure A6: Structural similarity index (SSIM) for Cascading Randomization on VGG16 network. Note that the legends have the same meaning as the column labels in Fig. 2.

The model reaches $99\%$ accuracy on the testing set. After training the model, we randomly sampled a subset of $2,000$ testing samples and used our method to derive explanations from the model predictions of samples in this subset. The dataset, model, and the descriptions of each feature can be found in `https://tinyurl.com/y9noqj6l`.

**Hyper-parameter choices.** The hyper-parameter choices of the proposed method on three datasets are shown in Table A1. In the table, $\ell$ is the index of the layer within the target model that is selected to generate the decoy images. The Lagrange multiplier $\lambda$ controls the weight of $\|F_\ell(\tilde{\mathbf{x}}) - F_\ell(\mathbf{x})\|_\infty$. The patch_size and stride control the size and the stride step of each decoy patch. $\tau$ is introduced by Eqn. 3 in Section A6. Note that we set the swappable patch size of SST and IDS data as 1, because their features may not have a strong local correlation. It should also be noted that we selected the swappable patch size of ImageNet data as the widely used convolutional kernel size 3 and stride size 1. We set the number of patches (masks) in each decoy $m$ as 100 for ImageNet, 1 for SST and IDS. When generating adversarial attack images, we applied the code released by the corresponding work (Ghorbani et al., 2017) and followed their default setup in our implementation.

## A11 SANITY CHECK FOR DECOY-ENHANCED SALIENCY MAPS

As suggested by Adebayo et al. (2018), any valid saliency methods should pass the sanity check in the sense that the saliency method should be dependent on the learned parameters of the predictive model, instead of edge or other generic feature detectors. We performed the model parameter randomization test (Adebayo et al., 2018) on the ImageNet dataset by comparing the output of the proposed saliency method on a pretrained VGG16 network with the output of the proposed saliency method on a weight-randomized VGG16 network. If the proposed saliency method indeed depends on the learned parameters of the model, it is expected that the outputs between the two cases differ substantially.

Following the cascading randomization strategy (Adebayo et al., 2018), the weights of pretrained VGG16 network are randomized from the top to bottom layers in a cascading fashion. This cascading randomization procedure is designed to destroy the learned weights successively. As illustrated in Fig. A5, the cascading randomization destroys the decoy-enhanced saliency maps combined with three existing saliency methods, qualitatively. The conclusion is also supported by quantitative comparison measured by the structural similarity index (SSIM), shown in Fig. A6.

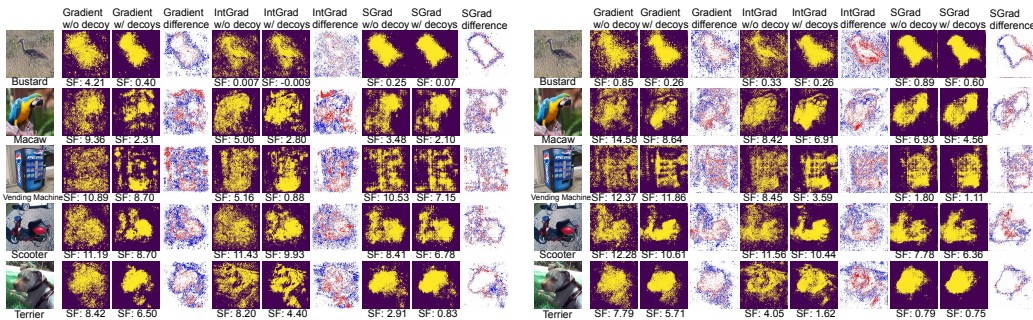

(a) Saliency maps generated on AlexNet.

(b) Saliency maps generated on ResNet.

Figure A7: Visualization of saliency maps under different CNN architectures. Here, the column labels are as same as those in Fig. 2. The difference figures share the same colorbar as those in Fig. 2.
Table A2: Quantitative comparison of our method and baselines on the network intrusion dataset. We report the means and standard errors of the fidelity scores.

| Salinecy method | Fidelity ($SF$) | | | | |
|---|---|---|---|---|---|
| | Without deocy | Decoys with range | Constant with range | Noise with range | Decoys with mean |
| Gradient | $1.80 \pm 0.39$ | $1.64 \pm 0.40$ | $1.68 \pm 0.40$ | $1.78 \pm 0.43$ | $2.04 \pm 0.40$ |
| IntegratedGrad | $1.68 \pm 0.39$ | $1.57 \pm 0.40$ | $1.68 \pm 0.44$ | $1.79 \pm 0.43$ | $2.19 \pm 0.39$ |
| SmoothGrad | $1.59 \pm 0.39$ | $1.57 \pm 0.40$ | $1.74 \pm 0.44$ | $1.73 \pm 0.44$ | $1.87 \pm 0.45$ |

## A12    APPLICABILITY TO OTHER CNN ARCHITECTURES

In addition to the VGG16 model, we generated saliency maps for AlexNet (Krizhevsky et al., 2012) and ResNet (He et al., 2016) trained from the ImageNet dataset. We visualize their saliency maps in Fig. A7. We observe that our method consistently outperforms the baseline methods, both quantitatively and qualitatively. Together with the results in Section 4, these results suggest that we can apply our decoy-enhanced saliency methods to various feed-forward network architectures and expect consistent performance.

## A13    PERFORMANCES ON THE NETWORK INTRUSION DATASET.

Rather than visualizing the saliency scores through heatmaps, we apply the following to compare the saliency scores obtained by different methods qualitatively. We ranked the features based on their saliency scores and compared the ranking obtained by the existing methods with that obtained by our decoy-enhanced method. "Minimum size of packet in forward direction", "Minimum length of a packet", "Minimum time between two packets sent in the forward direction" are ranked higher by our methods than the baselines. These features could capture the differences between benign and malicious traffics. This is because attackers usually tend to rapidly send small packages to discover the backdoors in the victim network system, while the benign users may send much larger packages with a longer interval between two packages. On the contrary, features that are not that useful for intrusion detection (*e.g.,* timestamp, Download and upload ratio) are wrongly pinpointed by the existing method. However, our methods correctly assign lower importance to these features. Table A2 shows the fidelity comparisons of different saliency methods. We can observe that our decoys-enhanced methods outperform the original saliency methods. These results show that our method could pinpoint more accurate features and achieve a higher fidelity than baselines. We also evaluated three alternatives used in Section 4: constant perturbation with range aggregation, noise perturbation with range aggregation, decoys generation with mean aggregation. The results in Table A2 are consistant with those in Fig. 2 and Fig. 3, i.e., our method outperforms these baselines. In summary, the results on this dataset align with those on the other datasets. This confirms our method's applicability to multilayer perceptrons.

## A14    DECOYS ON OTHER BASELINES.

In Section 4, we evaluated our methods on three state-of-the-art saliency methods. Recent research (Sturmfels et al., 2020; Hooker et al., 2019) suggests some variants that improve the perfor-

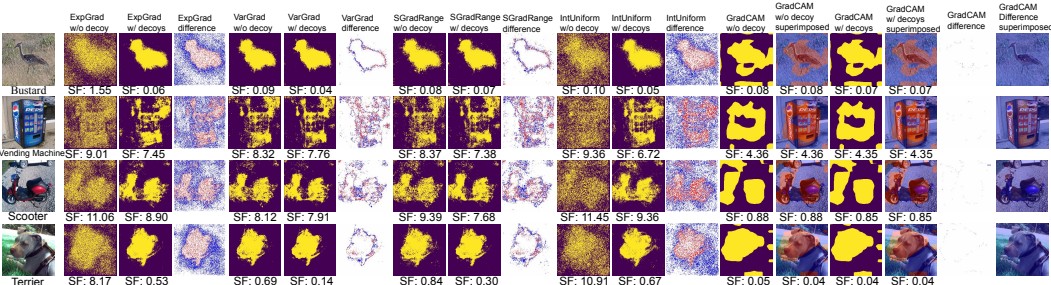

**Figure A8:** Visualization of saliency maps obtained by original saliency methods and our decoy-enhanced versions. "ExpGrad" refers to Expected Gradient, "SGradRage" stands for Smoothgrad with range aggregation, and "IntUniform" represents integrated gradient with uniform baseline. The difference figures share the same colorbar as those in Fig. 2.

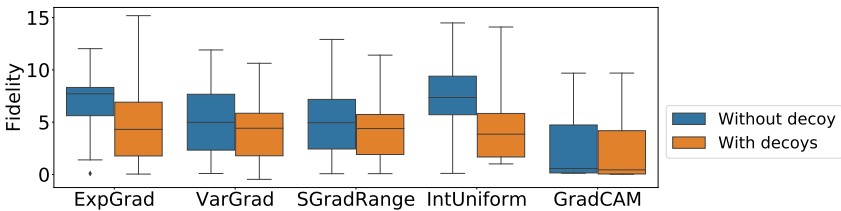

**Figure A9:** Fidelity comparision of saliency maps obtained by original saliency methods and our decoy-enhanced versions. "ExpGrad" refers to Expected Gradient, "SGradRage" stands for Smoothgrad with range aggregation, and "IntUniform" represents integrated gradient with uniform baseline (See Tab. A7 for more statistics about the performance differences).

mance of these baseline methods. Here, by using ImageNet data, we evaluate whether our decoy method could further improve these variants and another widely used saliency method. Specifically, we consider two variants of the integrated gradient: integrated gradient with uniform baseline (Sturmfels et al., 2020) and Expected Gradient (Sturmfels et al., 2020); two variants of the SmoothGrad: VarGrad (Hooker et al., 2019) and Smoothgrad with range aggregation; and one existing saliency method: Grad-CAM (Selvaraju et al., 2016). For the variants of the integrated gradient and Smooth-Grad, we kept the number of samples the same as the original version and used the default number suggested by existing works - 25 (See `https://github.com/PAIR-code/saliency`). We will investigate whether increasing the sample numbers improve the existing saliency methods' fidelity and robustness in future work.

Fig. A8 and Fig. A9 shows the qualitatively and quantitatively comparison of each method with/without decoys. As is depicted in Fig. A8, our method helps knock off the noises and improve the visual quality of the saliency maps. Fig. A9 further demonstrates the advantage of our method in explanation fidelity. Together with the results in Section 4, they demonstrate the generalizability of our technique to different saliency methods. Note that our method only imposes a minor improvement on Grad-CAM both qualitatively and quantitatively. As part of future work, we will explore how to customize our method for Grad-CAM and investigate the effectiveness of applying our technique to more saliency methods.

## A15 RUNTIME OF DECOY GENERATIONS

To evaluate the computational cost of our decoy generations, we carried out the run time comparison between optimizing one decoy and calculating three types of saliency methods, repeated $500$ times with respect to different patch masks. As illustrated in Fig. A10a, on average, optimizing one decoy is $62.3\%$ faster than the fastest vanilla gradient-based saliency method. For other methods, the optimization is even less expensive, in a relative sense.

As is mentioned in Section 3.4, our decoy generation, and the saliency map computation can be run parallelly in a batch mode. In the optimal case, where we have enough resources to compute each

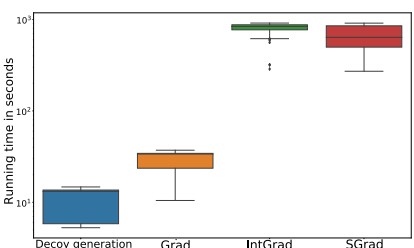
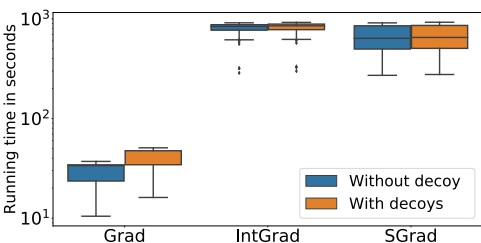

(a) Run time to optimize *one* decoy and calculate saliency map with the existing methods.

(b) Run time to compute saliency maps with and without optimizing *one* decoy.

Figure A10: Run time of decoy generation. The comparison is conducted in the same CPU/GPU to ensure fairness. Note that "Grad", "IntGrad", and "SGrad" stands for the vanilla gradient, the integrated gradient, and the SmoothGrad, respectively.

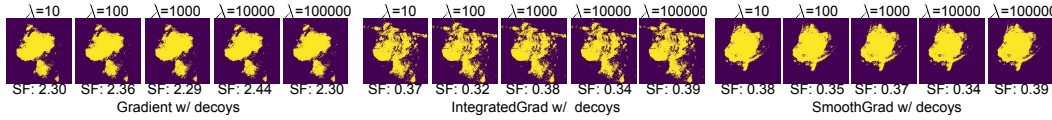

Figure A11: Visualization of saliency maps optimized using different initial $\lambda$.

decoy saliency map (*i.e.,* $E(\tilde{\mathbf{x}}^i; F)$) parallelly, the overall runtime of generating one decoy-enhanced saliency map is the total time of generating one decoy and computing one saliency map using the existing methods. Fig. A10b shows the comparisons between the optimal decoy-enhanced saliency map generation time and the original saliency map generation time across three saliency methods. As we can observe from the figure, our method introduces negligible computational overhead over the existing methods. Taking a step back, when the users have limited resources for running the decoy-enhanced saliencies in a fully parallel fashion. As is shown in Fig. 2, our method is not sensitive to the variations in the number of generated decoys. More specifically, Fig. 2 shows that we can obtain a decent performance by only solving 16 decoys on the ImageNet dataset. In the worst-case scenario where a user cannot run decoy generation in parallel, our method's computational overhead over the baselines is 24X for the Gradient approach, 16 X for the Integrated gradient method, and 16 X for the SmoothGrad method. In most cases, where users could afford partial parallel computing, this overhead will be decreased linearly with the available computational resources. For example, if a user has 4 GPUs, the overhead will drop to 6X for Gradient, 4X for Integrated gradient, and 4X for SmoothGrad. We argue that for an ensemble method, this overhead is acceptable. Besides, saliency generation is much lighter weight than training deep neural networks. Even with 4X~6X overheads, the time of computing saliency maps is still much less than network training. In addition, our method can be even faster on more powerful machines, which escalates the practicality of our method.

## A16 HYPER-PARAMETER SENSITIVITY

We also conduct experiments on the VGG16 to understand the impact of hyper-parameter choices on the performance of our optimization-based decoy generation method. Specifically, we focus on the choice of three hyper-parameters: network layer $\ell$, initial Lagrange multiplier $\lambda$, and patch size.

Accordingly, we first varied the value of $\ell$ for VGG16 and compared the differences of the generated decoy saliencies from the three aforementioned saliency methods. In particular, we set it to range from the first convolutional layer to the last pooling layer and demonstrate the generated decoy saliencies in Fig. A19. Note that according to our design, only the convolutional layers and the pooling layers can be used to generate decoy images. For each saliency method, Fig. A19 demonstrates that the decoy saliencies generated from different layers for the same image are of similar qualities. Fig. A19 also shows the mean and standard derivation of the $SF$ scores for each saliency method. These quantitative results also support the conclusion that our approach is not sensitive to the layer. This is likely because, as previous research has shown (Chan et al., 2015; Saxe et al., 2011), the final classification results of a DNN are not highly related to the hidden representations. As a result,

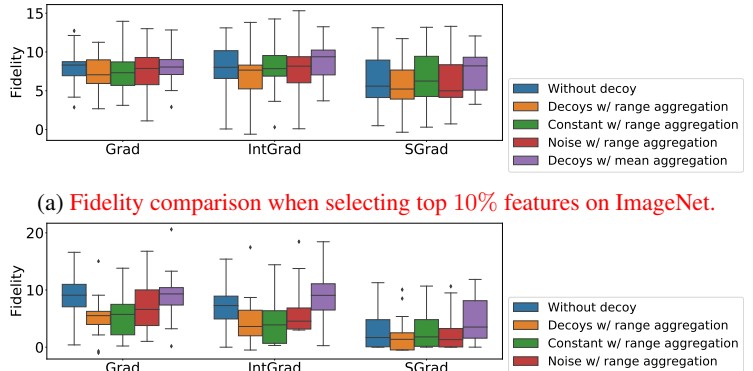

Figure A12: Visualization of saliency maps optimized using different patch size $P$.

(a) Fidelity comparison when selecting top 10% features on ImageNet.

(b) Fidelity comparison when selecting top 40% features on ImageNet.

Figure A13: Fidelity comparison of our methods and baselines under different choices of $K$ (See Tab. A8 and A9 for more statistics about the performance differences).

generating decoy saliencies for the same sample with the same label from different layers should yield similar results.

We also varied the initial Lagrange multiplier $\lambda$ to be $\left\{10^1, 10^2, 10^3, 10^4, 10^5\right\}$ and compared the differences of the generated decoy saliencies. Fig. A11 depicts the quantitative and qualitative comparison results. As shown in the figure, the different choices of initial $\lambda$ all produce similar saliency maps, indicating a negligible influence upon our method.

Then, we fixed $m$ and increased the patch size to be $\{3, 5, 7, 9, 11\}$ and showed the generated decoy saliencies in Fig. A12. The results show that varying the patch size within a certain range only imposes a negligible influence upon our method.

Recall that in Section 3.4, we mention that decoy masks are generated by sliding the swappable patch across a given input. With a given constant stride 1, the number of sliding windows is equal to $(\sqrt{d} - P + 1)^2$. In our implementation, to enable batch computing, we introduce $m$, which controls the number of sliding windows in each decoy. Then, the number of decoys is $2\left\lfloor(\sqrt{d} - P + 1)^2/m\right\rfloor$. Fig. A12 shows the results of fixing $m$ as 100 and varying $P$. In Fig. 2(C), we substantially varied both $P$ and $m$ and showed that our method is insensitive to the variations in the number of decoys $n$. Note that the box bars with the same color in Fig. 2(C) are drawn by fixing $P$ and varying $m$. Their slight difference indicates the robustness of our method in the variations of $m$.

The results in Fig. 2(C), A19, A11, and A12 indicate we can expect to obtain stable decoy saliencies when the hyper-parameters are subtly varied. This is a critical characteristic because users do not need to overly worry about setting very precise hyper-parameters to obtain a desired saliency map.

In addition to the hyper-parameters introduced by our methods, we also test the sensitivity of fidelity evaluation results to the choice of $K$ in the topK normalization. Specifically, we varied $K$ to select top 10% and 40% important features and redrawn the fidelity/sensitivity comparison figures in Fig. 2(B)/ Fig. 4 (B)~(D). The results in Fig. A13, A14, and A15 are aligned with those in Fig 2 and 4.

## A17 OBJECT LOCALIZATION

We compare our method and the vanilla gradient on the object localization task (Dabkowski & Gal, 2017; Fong & Vedaldi, 2017), where the model was trained with the class label only without access to any localization data. We carried out Imagenet ILSVRC'14 localization task (Russakovsky et al.,

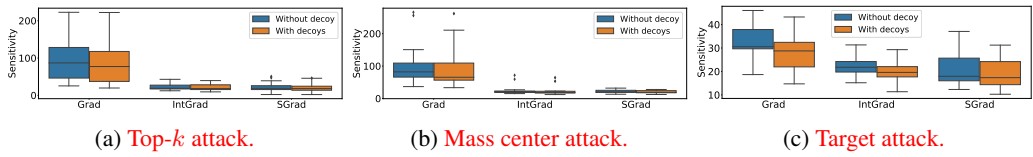

(a) Top-$k$ attack.       (b) Mass center attack.       (c) Target attack.

Figure A14: Sensitivity comparison when selecting top $10\%$ features on ImageNet (See Tab. A10 for more statistics about the performance differences).

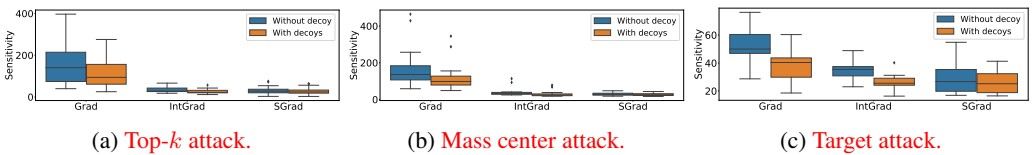

(a) Top-$k$ attack.       (b) Mass center attack.       (c) Target attack.

Figure A15: Sensitivity comparison when selecting top $40\%$ features on ImageNet (See Tab. A11 for more statistics about the performance differences).

Table A3: ImageNet localization accuracy on VGG16 network using different thresholding strategies.

| Accuracy | Value thresholding (0.25) | Energy thresholding (0.25) | Mean thresholding (0.25) |
|---|---|---|---|
| Gradient | 0.662 | 0.715 | 0.662 |
| Gradient w/ decoys | **0.722** | **0.723** | **0.665** |

2015) which contains 50K ImageNet validation images with annotated bounding boxes as ground truth. For each image, we first calculated the gradient-based saliency maps with and without using decoys, based on the pretrained model. Following the preprocessing steps suggested by Dabkowski & Gal (2017); Fong & Vedaldi (2017), we then obtained a bounding box from each calculated saliency maps based on certain thresholds. Specifically, we investigated three thresholding strategies suggested by Fong & Vedaldi (2017): value thresholding, energy thresholding, and mean thresholding. Following the evaluation protocol of Dabkowski & Gal (2017); Fong & Vedaldi (2017), we then computed the Intersect over Union (IoU) of the extracted box and the ground truth. If an IoU is greater than $0.5$, the corresponding box is marked as correct. Table A3 shows that decoy-enhanced saliency maps achieve higher accuracy than those of the vanilla gradient.

## A18  ADDITIONAL EXPERIMENTAL RESULTS

Fig. A17, Fig. A16, and Fig. A18 provide more results of the fidelity and robustness evaluation. These results are consistent with those shown in the Section 4.

## A19  STATISTICS OF THE PERFORMANCE DIFFERENCES

In section 4, Section A14, and Section A16, we varied the choice of $K$ in the top-$K$ normalizations, compared our method with each baseline approach, and showed the fidelity/sensitivity of each approach in the box-plots. To demonstrate the advantage of our method over the baselines, we further compared the fidelity/sensitivity difference between our method and the corresponding baseline approach. To be more specific, given two sets of fidelity/sensitivity scores ($s_{\text{our}}$ and $s_{\text{base}}$) obtained from our method and a baseline approach respectively, we first computed their difference, i.e., $diff = s_{\text{our}} - s_{\text{base}}$. Then, we conducted a statistical measure on the values of $diff$ by computing the mean, the standard error, and the $p$-value of the paired t-test. For the paired t-test, our null hypothesis is $H_0 : \mathbb{E}[diff] \geq 0$. This indicates that, if the value of $p$ is larger than a threshold, we cannot reject this null hypothesis, and have to conclude that our method cannot outperform the corresponding baseline approach. As we present in Table A4~Table A11, the overall experiment results align with those shown in the box plots, demonstrating the superiority of our method over the baselines.

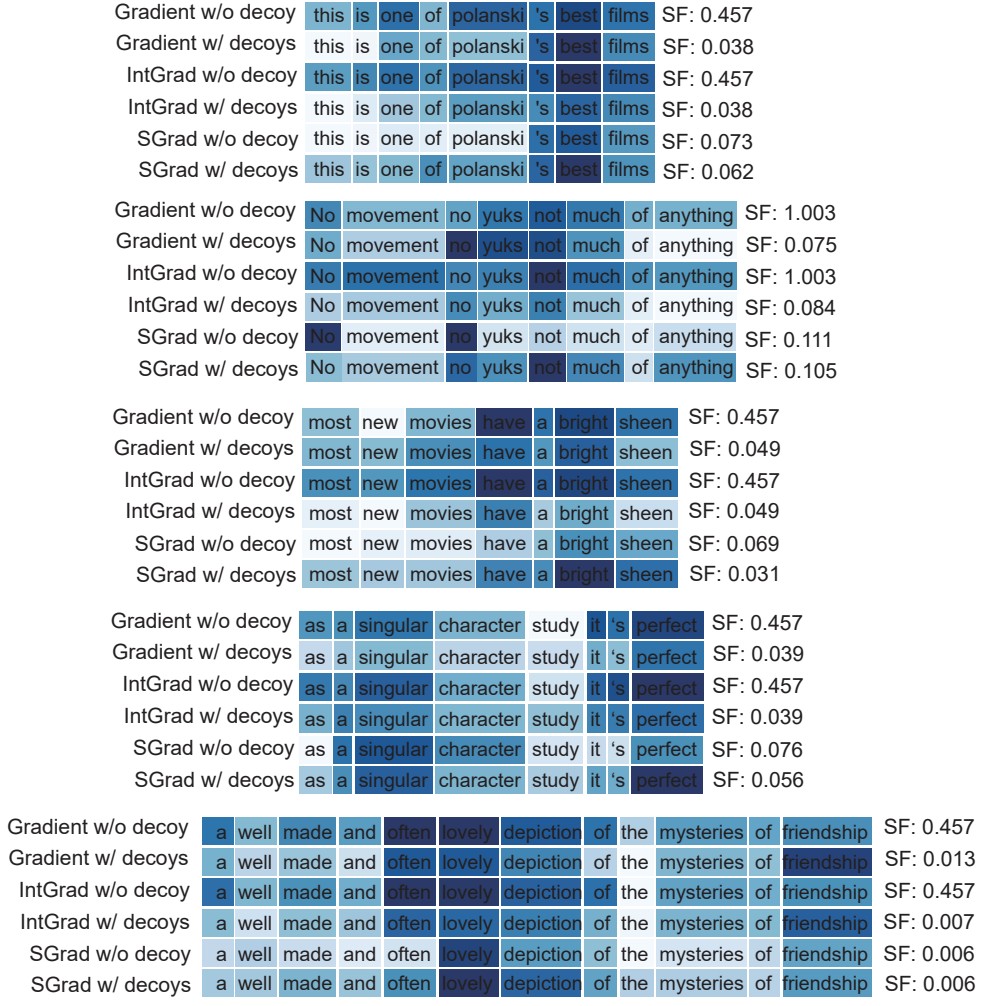

Figure A16: Visualization of saliency maps on the sentences in SST dataset. The row labels and colorbar are the same with those in Fig. 3(A).

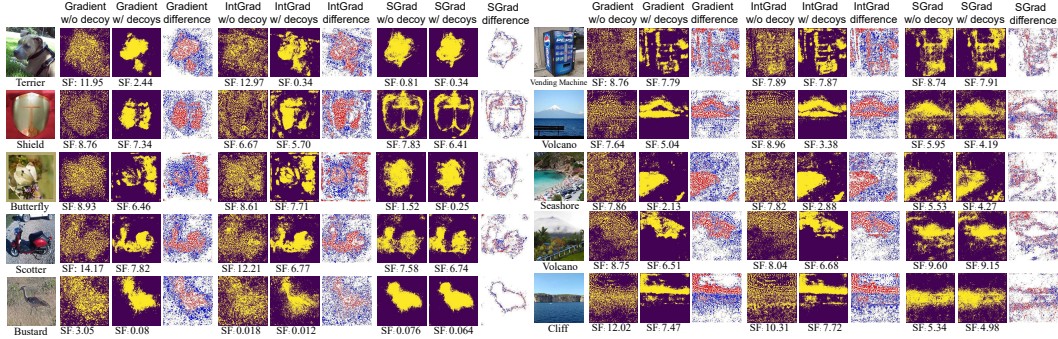

Figure A17: Visualization of saliency maps on the images in ImageNet dataset. The column labels and colorbar are the same with those in Fig. 2(A).

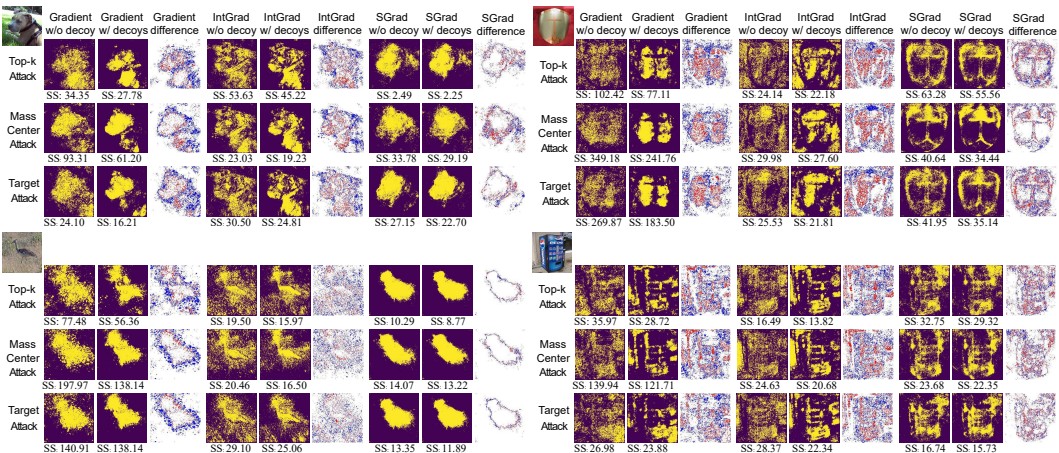

Figure A18: Visualization of saliency maps on the perturbed images generated by using three attacks in VGG16. The column labels are the same with those in Fig. 2(A).

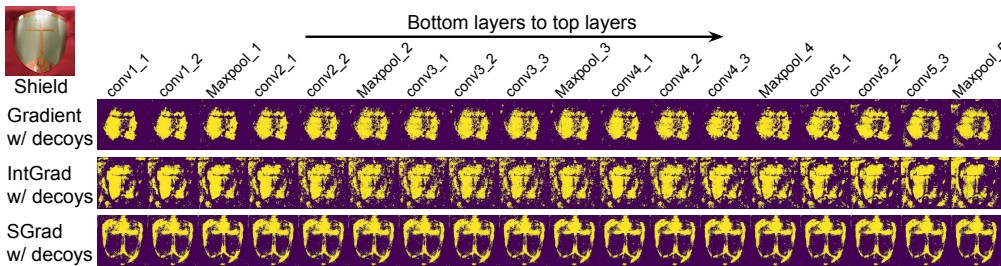

(a) The mean and standard derivation of $SF$ score for gradient, integrated gradient and SmoothGrad are: (10.23, 0.29), (10.37, 0.84), (9.34, 0.51).

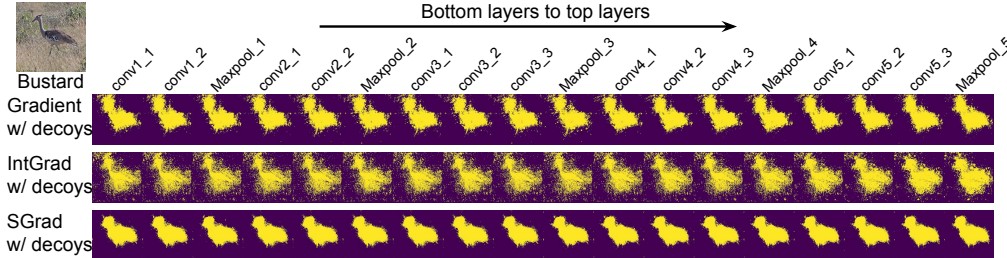

(b) The mean and standard derivation of $SF$ score for gradient, integrated gradient and SmoothGrad are: (0.07, 0.02), (0.01, 0.003), (0.06, 0.007).

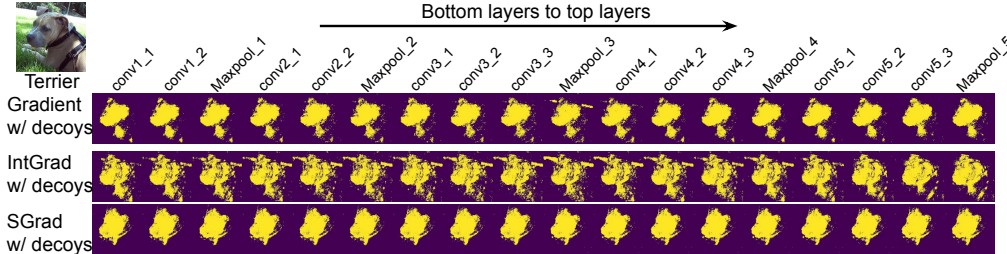

(c) The mean and standard derivation of $SF$ score for gradient, integrated gradient and SmoothGrad are: (2.15, 0.50), (0.97, 0.56), (0.19, 0.06).

Figure A19: Demonstrations of decoy-enhanced saliency maps generated from each convolutional and pooling layer in VGG16.

Table A4: Mean, standard error, and p-value of the difference in Fig. 2(B).

| Salinecy method | Without decoy | | Constant with range | | Noise with range | | Decoys with mean | |
|---|---|---|---|---|---|---|---|---|
| | Mean±Std | P-value | Mean±Std | P-value | Mean±Std | P-value | Mean±Std | P-value |
| Gradient | -1.61±3.24 | 0.014 | -1.26±2.29 | 0.009 | -0.51±1.74 | 0.093 | -1.80± 2.15 | < 0.001 |
| IntegratedGrad | -1.14±3.82 | 0.087 | -0.71±3.41 | 0.170 | -0.06±3.03 | 0.440 | -2.53± 2.25 | < 0.001 |
| SmoothGrad | -0.41±1.23 | 0.068 | -0.44±1.27 | 0.058 | -0.79±1.22 | 0.003 | -1.80± 2.65 | 0.002 |

Table A5: Mean, standard error, and P-value of the difference in Fig. 3(B).

| Salinecy method | Without decoy | | Constant with range | | Noise with range | | Decoys with mean | |
|---|---|---|---|---|---|---|---|---|
| | Mean±Std | P-value | Mean±Std | P-value | Mean±Std | P-value | Mean±Std | P-value |
| Gradient | -0.29±0.57 | < 0.001 | 0.003±0.09 | 0.921 | 0.003±0.09 | 0.912 | -0.17±0.51 | < 0.001 |
| IntegratedGrad | -0.12±0.56 | < 0.001 | 0.001±0.07 | 0.744 | -0.20±0.44 | < 0.001 | -0.09±0.52 | < 0.001 |
| SmoothGrad | -0.02±0.52 | 0.043 | -0.02±0.52 | 0.043 | -0.02±0.51 | 0.029 | 0.006±0.15 | 0.959 |

Table A6: Mean, standard error, and P-value of the difference in Fig. 4(B)∼(D).

| Attack | Gradient | | Integrated gradient | | SmoothGrad | |
|---|---|---|---|---|---|---|
| | Mean±Std | P-value | Mean±Std | P-value | Mean±Std | P-value |
| Top-k | -23.52 ± 57.02 | 0.008 | -3.89 ± 2.47 | < 0.001 | -2.32 ± 21.00 | 0.349 |
| Mass Center | -30.43± 25.48 | < 0.001 | -6.06 ± 4.56 | < 0.001 | -2.75 ± 1.85 | < 0.001 |
| Target | -7.66 ± 3.03 | < 0.001 | -4.77 ± 1.29 | < 0.001 | -2.81 ± 2.88 | 0.002 |

Table A7: Mean, standard error, and P-value of the difference in Fig. A9.

| ExpGrad | | VarGrad | | SGradRange | | IntUniform | | GradCAM | |
|---|---|---|---|---|---|---|---|---|---|
| Mean±Std | P-value | Mean±Std | P-value | Mean±Std | P-value | Mean±Std | P-value | Mean±Std | P-value |
| -2.26 ± 4.11 | 0.009 | -0.95 ± 1.18 | 0.001 | -0.66 ± 1.51 | 0.026 | -2.98 ± 3.18 | < 0.001 | -0.08 ± 0.25 | 0.121 |

Table A8: Mean, standard error, and p-value of the difference in Fig. A13a.

| Salinecy method | Without decoy | | Constant with range | | Noise with range | | Decoys with mean | |
|---|---|---|---|---|---|---|---|---|
| | Mean±Std | P-value | Mean±Std | P-value | Mean±Std | P-value | Mean±Std | P-value |
| Gradient | -0.87±2.13 | 0.034 | -0.30±1.21 | 0.126 | -0.29±1.01 | 0.100 | -0.91± 1.96 | 0.021 |
| IntegratedGrad | -1.39±2.29 | 0.005 | -1.31±1.64 | 0.001 | -0.79±1.50 | 0.011 | -2.02± 1.91 | < 0.001 |
| SmoothGrad | -0.79±0.97 | < 0.001 | -1.16±1.17 | < 0.001 | -0.58±1.01 | 0.007 | -1.76± 1.69 | < 0.001 |

Table A9: Mean, standard error, and P-value of the difference in Fig. A13b.

| Salinecy method | Without decoy | | Constant with range | | Noise with range | | Decoys with mean | |
|---|---|---|---|---|---|---|---|---|
| | Mean±Std | P-value | Mean±Std | P-value | Mean±Std | P-value | Mean±Std | P-value |
| Gradient | -3.26±3.88 | < 0.001 | -0.37±3.74 | 0.320 | -1.27±2.51 | 0.014 | -3.73± 2.75 | < 0.001 |
| IntegratedGrad | -2.31±3.70 | 0.004 | -0.21±2.99 | 0.374 | -1.87±3.08 | 0.005 | -4.33± 3.41 | < 0.001 |
| SmoothGrad | -0.94±1.21 | 0.001 | -0.94±1.09 | < 0.001 | -0.48±0.70 | 0.002 | -2.67± 2.92 | < 0.001 |

Table A10: Mean, standard error, and P-value of the difference in Fig. A14.

| Attack | Gradient | | Integrated gradient | | SmoothGrad | |
|---|---|---|---|---|---|---|
| | Mean±Std | P-value | Mean±Std | P-value | Mean±Std | P-value |
| Top-k | -8.04 ± 49.69 | 0.285 | -1.58 ± 1.75 | 0.003 | -1.34 ± 16.30 | 0.386 |
| Mass Center | -14.48± 15.68 | 0.003 | -2.98 ± 2.41 | < 0.001 | -1.87 ± 1.26 | < 0.001 |
| Target | -3.95 ± 2.42 | < 0.001 | -2.30 ± 1.06 | < 0.001 | -1.81 ± 1.96 | 0.003 |

Table A11: Mean, standard error, and P-value of the difference in Fig. A15.

| Attack | Gradient | | Integrated gradient | | SmoothGrad | |
|---|---|---|---|---|---|---|
| | Mean±Std | P-value | Mean±Std | P-value | Mean±Std | P-value |
| Top-k | -42.64 ± 76.08 | 0.032 | -8.37 ± 4.26 | < 0.001 | -2.81 ± 23.61 | 0.338 |
| Mass Center | -56.54± 38.27 | < 0.001 | -10.28±31.85 | 0.133 | -2.51 ± 1.49 | < 0.001 |
| Target | -13.09 ± 3.60 | < 0.001 | -8.29 ±2.08 | < 0.001 | -3.12 ± 3.69 | 0.005 |

