# OpenReview forum: "Decoy-enhanced Saliency Maps "
_ICLR.cc/2021/Conference — Reject_

### Official Review · AnonReviewer3 · 2020-10-28
**improving saliency maps with in-distribution noisy decoys**

**Rating:** 7
**Confidence:** 3

**Review:**

Summary: This paper introduces a technique to improve current gradient-based attribution methods by 1. generating perturbed data, i.e. decoys, that maintain the same feature maps as the original data and 2. an aggregation strategy of the saliency maps across the decoys. They provide theoretical as well as empirical support for their method. This paper show a great improvement over saliency maps and integrated gradients with a smaller gain for smoothgrad. This is well written and an interesting technique that could add to the repertoire of interpretability methods that exist.

Comments:
* One questions that arose while reading: why is there a significant boost with range aggregation compared to mean aggregation? What is the intuition here?
* Visually, smoothgrad (without decoys) seems to make a big improvement over saliency maps and integrated gradients — almost as much as decoy-enchanced saliency maps. Smoothgrad perturbations are drawn from Gaussian noise, not optimized, with a mean aggregation. The performance gain of smoothgrad seems to be better than the decoys with mean aggregation (Fig. 1). Certainly the decoys with range aggregation improve Grad (saliency maps). Does it make sense to try smoothgrad with the range aggregation strategy to see if it improves its performance?  This experiment can help isolate which attributes are more noteworthy, the decoy generation method itself or the aggregation strategy.  Also, since smoothgrad can be applied to integrated gradients, a similar comparison should be made for integrated gradients as well.
* The fidelity metric is a way to quantitatively compare the attribution methods, but it is unclear (at least visually) how sensitive the score is. For instance, in Fig. 1, the picture of the alps has a SF score of 7.57 for SGrad_Decoys, while the the volcano image has an SF score of 0.72. Moreover, the intgrad_decoys for the volcano has an SF score of 0.36, but it seems to be capturing the base of the mountain in addition to the sky. Are there certain aspects of the saliency maps that boosts/biases this fidelity score? A brief mention of the limitations of this SF score would be helpful to put this in context for a reader.
* The comparison between grad and intgrad with deocys is visually very clear. However, the difference between sgrad and sgrad+decoys is much less pronouced. Perhaps a difference plot would be better at revealing what the decoy methods are able to better capture.
* The robustness to adversarial attacks on images is not as convincing as the other aspects of the story. In Ghorbani et al. 2017, they demonstrate that adversarial attacks can maintain the same predictions but lead to very different attribution maps.  Each attack that is explored here (Top-k, mass center, and target) largely leads to the same feature importance maps as the original image without any adversarial attacks (see Fig. 3a). This may be why the sensitivity (while consistently lower) is very similar to the feature importance maps without decoys.
* On a smaller note, this paper mentions that the three hyperparameters are swappable feature size, network layer, and initial lagrange multiplier. They never mention lagrange multiplier in the main text — it is first introduced in A6.  It should probably be mentioned in the main text for clarity.

---

> ### Author Response · Authors · 2020-11-22
> **[Responses to AnonReviewer3]-Responded to the reviewer's questions; Added the required baseline; Updated visualization figures and main text by following the reviewer's suggestions.**
>
> We would like to thank the reviewer for reviewing our paper. A point-by-point response is shown below.
>
> 1. The reviewer questioned the motivation for using range aggregation instead of mean aggregation. To respond to this question, we first show that range aggregation performs better than mean aggregation in Fig. 2 & 3, and Table 2A. Second, parallel work in statistics also demonstrates the effectiveness of using a range-based metric to control the false discovery rate in variable selection[1].
>
> 2. The reviewer suggested that we could use SmoothGrad with the range aggregation strategy. Following this suggestion, we conducted several experiments and added Fig. A8 & A9. The results in both figures indicate that SmoothGrad with range aggregation (denoted as SGradRange) does not outperform the original SmoothGrad method, whereas our method boosts its performance.
>
> 3. The reviewer asked about potential biases and limitations of the calibrated fidelity metric. This question has also been asked by reviewer 2. To respond to this question, in our new submission, we changed our preprocessing and evaluation process and then updated all the experiment results accordingly. Specifically, first, to avoid the bias introduced by saliency values, we changed our normalization method to retaining only the top-20% features ranked by each saliency method (Note that we also vary the value of K. In Section A16, we showed that the choice of K does not change our conclusion). Second, to rule out the bias introduced by the calibration term (the denominator), we deleted it and switched back to the commonly used fidelity metric shown in Eqn. (8). These changes could reduce the bias introduced by the normalization and fidelity metric. We revised Section 5 to point out the limitation of the new metric. That is, $E(\mathbf{x} ; F^c) \circ \mathbf{x}$ may be out-of-distribution, causing poor performance of the classifier. We also stated in Section 5 that we will investigate more rigorous metrics and use other benchmark datasets(e.g., BAM[2]) for evaluation.
>
> 4. We followed the reviewer’s suggestion and visualized the difference of saliency methods with and without using our decoy method in the new submission.
>
> 5. The reviewer questioned the effectiveness of the adversarial attack. In Section A10, we made changes and stated that we applied the code released by the corresponding work and followed their default setup in our implementation.
>
> 6. We followed the reviewer's suggestion and moved the Lagrange multiplier, the parameter introduced in the implementation details, to the main text.
>
>
> [1]“Neural Gaussian Mirror for Controlled Feature Selection in Neural Networks”.
> [2]“Benchmarking Attribution Methods with Relative Feature Importance’’.

---

### Official Review · AnonReviewer2 · 2020-10-29
**Interesting method, but some concerns regarding the benchmarking/runtime**

**Rating:** 6
**Confidence:** 4

**Review:**

(I have updated my review to raise my reviewer score by two points after discussion with the authors; this is still a preliminary evaluation as I have not discussed the paper with the other reviewers)

--

SUMMARY OF METHOD

The paper discusses an approach to create "decoys", which are slightly altered versions of the input that preserve the activations of an intermediate layer. The proposed "decoy enhancement" consists of applying a saliency method to calculate importance scores for the decoys, and then subtracting the maximum saliency across decoys from the minimum saliency. The authors benchmark their method on several datasets and argue that their saliency maps are superior according to the fidelity metric (Dabkowski & Gal, 2017). They also demonstrate that the decoy-enhanced maps are more robust to adversarial attacks.

STRENGTHS

- This approach of enhancing saliency maps is very unique compared to other methods in the literature. The generation of decoys could be useful in other contexts (e.g. to understand which sets of inputs can compensate for or "buffer" each other).
- The authors have benchmarked their method on several types of data - images, the Stanford sentiment treebank, and network intrusion detection.
- The authors have included several valuable experimental analyses, such as making sure the method is robust to the choice of hyperparameters, performing sanity checks to make sure the saliency maps respond to cascading randomization (a sanity check from Adebayo et al.), investigating different baselines for integrated gradients, comparing the decoys to a constant perturbation, etc.
- To me, the authors have made the case that the maps are more robust to adversarial attacks (caveat: I am not an expert on adversarial attacks)

WEAKNESSES

(1) The biggest weakness of this method would be the runtime to compute the saliency maps on the decoys. The authors generate $2n$ decoys, where $n$ is the number of unique masks. The number of masks is determined by the patch size $K$, and the authors write (in section A16 in the supplement) "given a constant stride 1, the number of decoys $n$ is unique to the patch size $K$. Specifically, $n$ is inversely proportional to $K$". The claim that $n$ is inversely proportional to $K$ is a little confusing given that the stride is 1 - I would expect the number of unique masks to be $(\text{image_len} - $K$ + 1)^2$, but the phrase "inversely proportional" implies that $n$ is proportional to $1/K$. Assuming that the authors meant  $(\text{image_len} - $K$ + 1)^2$ (which is consistent with a stride of 1) - this could potentially be a very large number of decoys per image, given that the patch sizes $K$ are fairly small ({3, 5, 7, 9, 11} according to section A16 in the supplement) - e.g. the ImageNet images are resized to 227 x 227 (as per section A10 in the supplement). The authors say in section A15 of the supplement that the "saliency map computation can be run parallelly in batch mode", but regular saliency maps can also be computed in batches on a GPU, so I am confused how the authors claim that their method  "introduces about 2X ~ 5X overhead over the existing saliency methods" when it seems like the number of decoys per image is much larger than 5. Can the authors explicitly state how many decoys are used for particular tasks, and (if the number is larger than 5) how they reconcile this with the claim of "2X - 5X overhead"? It would additionally be good to clarify whether the interpretation improvements from having the decoys are greater than what one would achieve by instead investing the equivalent computation time in other techniques (e.g. more samples for SmoothGrad or ExpectedGradients - have SmoothGrad and ExpectedGradients saturated in the number of samples?).

(2) I have a set of concerns pertaining to the evaluation using the fidelity metric. The authors define a normalized saliency score as follows: "First, we follow the existing methods (Simonyan et al., 2013) and compute the absolute saliency scores...to avoid outlier features with extremely high saliency values leading to almost zero saliency scores for the other features, we then winsorized outlier saliency values to a relatively high value (the 95th percentile)...before linearly scaling to the range [0,1]". They then define the fidelity metric as $- \log \frac{F^c (E(x; F^c) \odot x) }{F^c (\bar{E(x; F^c}) \odot x) }$, where $c$ is the predicted class of the input. The authors write "by viewing the saliency score of the feature as its contribution to the predicted class, a good saliency method will weight important features more highly than less important ones and thus give rise to higher predicted class scores and lower metric values. Note that we subtract the mean saliency $\bar{E(x; F^c)}$ to eliminate the influence of bias in $E(x; F)$ and exclude trivial cases such as $E(x; F^c) = 1$". Here are my concerns:

2a) In Figure 1, the fidelity scores are predominantly all positive - this implies that $F^c (E(x; F^c)$ was *lower* than $F^c (\bar{E(x; F^c}) \odot x)$, which is concerning - the average saliency map across all images was better than a saliency map that was generated specifically for the example at hand? To me, this suggests that the fidelity metric is not working as intended on image data. My guess is that there is a lot of *high-frequency noise* in the saliency maps, which is causing $E(x; F^c) \odot x$ to be out-of-distribution, and that is the reason $F^c(E(x; F^c) \odot x)$ is so poor. According to this hypothesis, I suspect the authors may find that simply *smoothing* the saliency map scores would achieve better results via the fidelity metric. The authors should thus include simple smoothing as a baseline. I note that the fidelity metric values are predominantly negative for the SST dataset (as shown in Figure 2c), which a sign that the fidelity metric makes sense there; however, in Table A2 (pertaining to the network intrusion dataset), the fidelity scores are once again predominantly positive (also Grad performs better than IntGrad and SGrad, which is unexpected).

2b) The authors acknowledge that extreme values can result in near-zero importance for many features in the normalized saliency maps and propose clipping as a way to address this; however, even with clipping, the problem could still persist; if the 95th percentile of a saliency method is more on the extreme side, then the resulting masks would have a **lower L1 norm after the linear scaling normalization** - thus, saliency maps that had less extreme variation in the scores would get an unfair advantage in the evaluation (because they would be able to retain a larger portion of the original image after applying the mask $E(x; F^c)$). It is not clear to me that subtracting the mask created by the mean saliency would be enough to correct for this potential advantage; the rigorous way to correct for it would be to retain something like the top $k$ positions as ranked by a particular saliency method (with sign information included for the methods that provide them), or to find some other way of ensuring that the L1 norm of the masks were comparable.

2c) As alluded to in 2b, the approach of taking the absolute value of the saliency scores would handicap methods where the sign information is valuable. Given that the authors are specifically trying to identify regions that are important for the target class, they should retain the sign information for the methods that provide them (in conjunction with switching to a rank-based masking strategy described above). I recognize that Simonyan et al., 2013 discarded the sign information, but they were not trying to optimize the fidelity metric. I also recognize that the proposed decoy-enhanced saliency scores are always positive, but this is a shortcoming of the decoy-enhanced saliency scores (in that they don't distinguish regions that are contributing positively to a class from regions that are contributing negatively to the class); I don't think that's a good enough reason to handicap methods that are providing valuable information in the sign of the saliency.

2d) (this point is lower priority) Again regarding the benchmarking on images: it has frequently been observed that pixel-level attribution methods do not perform very well on image data (e.g. they tend to be outperformed by Grad-CAM, which computes the attributions at a higher conv layer). Accordingly, to make a compelling case that the proposed method can be helpful in practice, the authors should show that the decoy enhancement helps even when the saliency maps are computed at some higher convolutional layer (rather than computed at the pixel level). Alternatively, they could add Grad-CAM (or a related method - see [1]) as a baseline. A suggestion regarding benchmarking on image data: the authors could consider using the BAM dataset (https://github.com/google-research-datasets/bam) to sidestep the issues with the fidelity metric altogether.

(3) In section A7 of the supplement, the authors write "from the above equation, we can see that, for a linear model, the linearity zeroes out the gradients of the decoys, causing the ill-defined behavior of our method. This does not dilute the significance of our method because linear models are self-explainable and do not need extra post-hoc explanation". I think this is a bit misleading (either that, or it could use more clarification); the equation in question is $\nabla_{\tilde{\boldsymbol{x} }} F^c(\tilde{\boldsymbol{x}})^T \Delta \approx - \frac{1}{2} \Delta^T \boldsymbol{H}_{ \tilde{\boldsymbol{x} } } \Delta$, where $\Delta = \boldsymbol{x} - \tilde{ \boldsymbol{x} }$ and $ \tilde{ \boldsymbol{x} }$ denotes the decoy of $x$. To me, this equation just seems to convey that decoys are constructed such that the change in the output due to first-order effects (the left-hand side) is approximately balanced by the change in the output caused by second-order effects (the right-hand side). If the right-hand side were zero (as is the case in a linear model), then that would just mean the first-order effects would have to be such that they canceled out - why does this make the method's behavior "ill-defined"? However, more to the point: it seems clear that the decoy-construction process will only perturb a feature if they effect of this perturbation can be cancelled out by perturbing other features. But *a feature can be important even if the effect of perturbing the feature is not easily cancelled out by perturbing other features* - particularly given that the decoys are constructed in order to preserve the representation at some intermediate hidden layer. Also, regarding the assertion that "linear models are self-explainable and do not need extra post hoc explanation" - a more complex model is still capable of learning linear effects for some of its input features, and it would not be obvious that the complex model has learned these linear effects without some post-hoc explanation.

(4) Since these decoy-enhanced saliency maps look at $\max(\tilde{E}) - \min(\tilde{E})$, where $\tilde{E}$ is a population of saliency scores, it seems appropriate to benchmark against a method that also looks at the variation across a population of saliency scores - e.g. the VarGrad method, which was one of the top-performing methods in https://arxiv.org/abs/1806.10758. Also, while I appreciate that the authors added a baseline of the "constant perturbation" decoy, I would be interested to see a similar baseline that consists of random perturbations that are of comparable magnitude to the perturbations made by decoys.

MINOR

- Why do the authors write (in the introduction, when discussing their proposed method): "By design, this score naturally offsets the impact of gradient saturation" - if the gradients on a particular set of inputs is zero, would the decoy-generation process perturb those inputs? My understanding is no, since the decoy generation proceeds via gradient descent. In what sense is the saturation problem addressed?
- In table A3, how many samples were used for ExpectedGradients?
- It would be good if the authors clarified what $\delta_i$ is in Proposition 1 in the main text itself, without the reader having to refer to the supplement
- There are a few typos/grammatical errors - "it exhibits ill-defined on linear models" on page 4, "can obtain where" in Proposition 1, "obtain, we can further" in the subsequent paragraph.
- I did not read the supplementary proof of the adversarial robustness in detail (I am also not an expert on these adversarial attacks), so one high-level question I have for the authors is whether we expect *any* method that is based on an ensemble of saliency maps to be more robust to adversarial perturbations compared to the original (non-ensembled) saliency maps, and if so, is the proposed decoy-enhanced approach *more* robust than what would be provided by devoting the equivalent computational resources to drawing more samples for an ensemble (say with a method like VarGrad)?

SUMMARY OF RATING

I think the concept of using decoys to enhance saliency scores is unique, and the authors have clearly invested considerable effort in the work. However, the concerns regarding the benchmarking (point (2) under weaknesses) mean that I hesitate to fully believe the empirical improvements (the results of course look aesthetically pleasing to the human eye, but we as a field have learned our lesson that aesthetically pleasing results can be deceiving). If the authors can make the benchmarking more compelling, I would be willing to revise my score towards acceptance. Overall, my current feeling is that the decoy generation process proposed by the authors may turn out to have good uses (e.g. in identifying sets of inputs that can compensate for each other), but I'm not yet convinced that the application presented here of enhancing saliency maps is the right fit for the method (particularly when coupled with the runtime concerns in point (1) and the potential failure cases in point (3) where important features might not be highlighted).

[1] https://openaccess.thecvf.com/content_CVPR_2020/papers/Rebuffi_There_and_Back_Again_Revisiting_Backpropagation_Saliency_Methods_CVPR_2020_paper.pdf

---

> ### Author Response · Authors · 2020-11-22
> **[Responses to AnonReviewer2]-Responded to the reviewer's questions; Reran the experiments by following the reviewer's suggestion on the normalization method and evaluation metric; Added the required baselines.**
>
> We thank the reviewer for the constructive comments and respond as follows.
>
> 1. The reviewer asks whether the runtime is prohibitively long. To answer this question, our new submission includes several changes. First, Section 3.4 clarifies that multiple decoy masks can be aggregated into a decoy sample and optimized jointly. This reduces the runtime significantly. Second, Section 3.4 clarifies how we compute the decoy sample size $2n$. The decoy sample size depends on the patch size $P$ and the number of masks $m$ in one decoy sample. To ensure a low runtime overhead, we can control $m$ and $P$, reduce the decoy size, and thus lower the runtime overhead. Third, we added Fig. 2(C), showing that a smaller $n$ (e.g., n=16) can achieve decent interpretation fidelity. Section A15 further shows that the time required to generate one decoy is small compared to existing saliency methods. This further indicates that our method can improve on existing methods without too much computational overhead (more details can be found in Section 5). Last, Section A15 now includes a paragraph pointing out that runtime could be reduced by using GPUs to parallelize the decoy computation.
>
> 2. To avoid the bias introduced by saliency values, the reviewer suggested retaining and binarizing the top-K important features as the preprocessing of saliency maps. We followed this suggestion, retained the top-20% features, and updated all the experiment results. (Note that we also experimented with the effect of K in Section A16).
>
> 3. Our prior submission stated that our calibrated fidelity metric is designed to knock off the pathological case where every feature's saliency value equals 1. The reviewer pointed out a flaw in this metric. We agree with this critique and made corresponding changes. specifically, we now use top-K normalization in our experiments. This change naturally ensures that our method will no longer suffer from pathological cases. As such, our new submission removes the calibration term and switches back to the commonly-used fidelity metric in Eqn. (8). We agree with the reviewer that the commonly-used fidelity metric could still cause the out-of-distribution problem. Section 5 states that we will explore alternative metrics and other benchmarks (e.g., BAM[1]) as part of our future work.
>
> 4. Following the reviewer’s suggestion, we updated our experimental results by choosing the top-K features from the raw saliency values rather than the absolute values.
>
> 5. We followed the reviewer’s suggestion and applied decoys to Grad-CAM. The newly added Figure A8 & A9 show that combining decoys with Grad-CAM improves fidelity marginally. Our new submission states that we will explore methods to address this issue in the future.
>
> 6. The reviewer asked why linear models make our method's behavior ill-defined. As shown in Eqn. 6 & 14, our method outputs a zero score for all features on linear models. We revised Section A7 to clarify our method's problem on linear models and delete the incorrect statement pinpointed by the reviewer.
>
> 7. The reviewer stated that ‘’a feature can be important even if the effect of perturbing the feature is not easily canceled out by perturbing other features’’. We hypothesize that a single pixel generally has a joint effect with its neighbor pixels in order for the model to capture meaningful patterns such as edges, texture, etc. This indicates that a single pixel has room to fluctuate without influencing the prediction. When the effect of perturbing a single pixel cannot be canceled out by perturbing its neighbors, we hypothesize that this pixel might be under adversarial manipulation. In this case, our method will assign a low saliency score to this pixel and offset the effect of adversarial alteration.
>
> 8. We followed the reviewer’s suggestion and applied decoys to VarGrad. The newly added Fig. A8 & A9 show that decoys can improve VarGrad.
>
> 9. We followed the reviewer’s suggestion and added noise perturbation as a baseline of our evaluation. The newly added Fig. 2 & 3 and Table A2 show that noise perturbation cannot outperform our method.
>
> 10. The reviewer asked  how our design addresses gradient saturation. Our design addresses it in two ways. First, similar to the IntegratedGrad and SmoothGrad, we move features from the saturation region by adding perturbation. Second, when solving decoys, before applying the gradient descent, we add a small perturbation to the input via random initialization by following the insight of SmoothGrad. This helps avoid the zero gradients of saturated inputs.
>
> 11. In section A14, we followed the reviewer's suggestion and added our choice of the sample number in ExpGrad. We revised A14 and stated that our future work will investigate whether the increase in the sample number will improve the existing saliency methods' fidelity and robustness.
>
> [1]“Benchmarking Attribution Methods with Relative Feature Importance’’.

---

> > ### Comment · AnonReviewer2 · 2020-11-24
> > **Thanks for addressing concerns**
> >
> > Thanks for addressing the concerns and for the clarifications. I will spend more time looking over the changes in detail but wanted to send some quick follow-up questions in case the authors have a chance to respond before the window for conversation expires (but I understand there is not much time left)
> >
> > 1) The updated performance plots seem to have bars that overlap each other a lot more. Are the authors able to put p-values on the differences, particularly for box plots that seem to overlap a lot? I will also look closer to see if this is mentioned in the text. I'm assuming in good faith that the differences are significant, but it would help if this were confirmed.
> >
> > 2) I appreciate that the authors say they will try different benchmarks in future work to address the out-of-distribution issue. I'm trying to assess whether the out-of-distribution problem is likely to be playing a significant role in the current results. Could the authors comment on whether they found that the  method tends to do better (according to the fidelity metric) than simply using the average saliency map? (I know the authors said there is no need to use the average saliency map for the calibration term anymore, but I am still curious whether the average saliency map does better, which was a concern in my original comment for the ImageNet and Network Intrusion datasets).

---

> > > ### Author Response · Authors · 2020-11-24
> > > **[Responses to AnonReviewer2's new comments (Part 2/2)] - Ran the suggested experiments and presented the experimental results.**
> > >
> > > 2. Under the evaluation fidelity metric, the reviewer asked whether our method outperforms the method that simply uses the average saliency map. With regard to the question, we have the following three understanding with regard to the form of the “average saliency map”.
> > >
> > > 1). Average of min-max normalized saliencies($-\log F^{c}(\bar{E}(\mathbf{x} ; F^c)’ \circ \mathbf{x} )$). This form is mentioned in our prior submission. In this form, each input feature is weighted by the average value of the normalized saliency map given the corresponding sample as input. Note that we compute the average saliencies on the saliency maps normalized by the previous min-max normalization method (We clipped the 95th percentile of the raw saliency scores rather than the absolute values and then conducted the min-max normalization).
> > >
> > > 2). Average of top-K normalized saliencies($-\log F^{c}(\bar{E}(\mathbf{x} ; F^c) \circ \mathbf{x} )$). This form is almost the same as the above form, except that we computed the average saliencies on the saliency maps normalized by the new top-K normalization method (We binarized the top-K ranked features as 1 and kept the rest features as 0).
> > >
> > > 3). Average fidelity($-\log (\bar{F^{c}}(E(\mathbf{x} ; F^c) \circ \mathbf{x} )$). This form better aligns with the top-K normalization used in our revised submission. In this form, we shuffle the positions of the top-K features multiple times and compute the average fidelity for each shuffled saliency map.
> > >
> > > Since we are not sure which particular form of the average saliency map the reviewer is referring to, we compared our method with all of them on three selected datasets and put the results in the following doc: https://tinyurl.com/y4jxmkvo. First, as we can observe from the results, our method consistently outperforms the “Average fidelity” on the three datasets, indicating our method indeed captures the feature importances. Second, we can also observe that our method beats the “Average of min-max normalized saliencies” and “Average of top-K normalized saliencies” on SST and IDS. Finally,  “Average of min-max normalized saliencies” and “Average of top-K normalized saliencies” achieve lower fidelity on the ImageNet than our method. We hypothesize there are two reasons behind this observation. First, just like what the reviewer pointed out, ($E(\mathbf{x} ; F^c) \circ \mathbf{x}$) goes out of distribution by a larger margin than $(\bar{E}(\mathbf{x} ; F^c) \circ \mathbf{x})$. Therefore, our method has worse prediction results than the  “Average of min-max normalized saliencies” and “Average of top-K normalized saliencies”.  Since both  “Average of min-max normalized saliencies” and “Average of top-K normalized saliencies” use a semi-transparent image as the input, this result may imply that, for the ImageNet dataset, the semi-transparent images are better immune to the out-of-distribution problem than the images with only a small proportion of the pixels retained. This confirms the reviewer’s concern about the potential problem of the fidelity metric. As we mentioned in the previous response, we will investigate this problem in our future work. Second, this may also be caused by an unfair comparison. For the min-max normalization, every element in an average saliency of a given saliency map equals the mean value of the saliency scores in the given map. For the top-K normalization, every element of the average saliency across all given saliency maps equals the percentage of the selected features (In our case, the value 0.2). For both normalization methods, the average saliency still keeps all the features while our method only preserves the top 20% features. We hypothesize that the variation in the number of preserved features can also cause the worse fidelity of our method than the  “Average of min-max normalized saliencies” and “Average of top-K normalized saliencies”. In the condensation of this, in our experiment, we kept the same K for all the methods and tried to enable a fair comparison. Because we are not sure which baseline to use, we didn’t include the average saliency experiments in the revised paper. We would like to ask for the reviewer’s suggestion and include the results in our next version.

---

> > > > ### Comment · AnonReviewer2 · 2020-11-25
> > > > **Thanks for the updates**
> > > >
> > > > Thank you for getting back to me; in light of these updates, I'm happy to raise my score by two points, placing the paper above the acceptance threshold (that said, this is still a preliminary score as I have not discussed the paper with other reviewers yet). Even though the p values show that the proposed method occasionally does not significantly outperform the baselines, it seems to generally perform better.
> > > >
> > > > Regarding the average saliency map and the concern of "the average saliency still keeps all the features while our method only preserves the top 20% features": I was actually envisioning the authors would first take the average of the raw saliency scores across all examples and then perform the top-K normalization at the very end; my understanding is that this might result in a good head-to-head comparison and address the concern of preserving different numbers of features?
> > > >
> > > > For the final submitted paper, given the concerns about out-of-distribution artifacts for the imagenet fidelity evaluation, I think it would be better (in terms of ensuring that the reader leaves with an accurate take-home message) if the imagenet results were prioritized less (unless the out-of-distribution concerns are resolved in a different way). No need to update the current manuscript right now to reflect; I'm just saying that would be good revisit this for the camera-ready version if the paper is accepted.

---

> > > > > ### Author Response · Authors · 2020-11-25
> > > > > **[Response to AnonReviewer2's latest comments]-Ran the experiments about the new way of computing average saliency maps and reported the results.**
> > > > >
> > > > > We thank the reviewer for the prompt response. We really appreciate the constructive feedback and the reviewer’s kindness. In the following, we respond to the reviewer’s new comments.
> > > > > 1. Regarding the way to compute the average saliency map, our understanding is as follows. Given a raw saliency map, this method first computes an average saliency map in which every feature equals the mean value of the saliency scores in this map. Then, it performs top-K normalization with the choice of K equal to 20\% (the value we set up in our new experiment).  With this understanding, we just conducted a new experiment and presented the experimental result at https://tinyurl.com/y4jxmkvo. As we can observe, our method outperforms this average saliency approach, demonstrating a lower fidelity score.
> > > > >
> > > > > 2. We totally agreed with the reviewer. Without addressing the out-of-distribution concern, the ImageNet results should have a lower priority. In the next version, by using the average saliency experiment results (show at https://tinyurl.com/y4jxmkvo), we will first demonstrate and clarify that the fidelity metric suffers from the out-of-distribution problem on the ImageNet dataset. If we could solve the problem before the camera-ready deadline (if the paper could be accepted), we will address this concern and update the results accordingly. Otherwise, we will follow the reviewer’s suggestion and lower the priority of the ImageNet results. Once again, we thank the reviewer for providing us with constructive feedback and helping us put the paper in better shape.

---

> > > ### Author Response · Authors · 2020-11-24
> > > **[Responses to AnonReviewer2's new comments (Part 1/2)] - Ran the suggested experiments and revised the paper accordingly.**
> > >
> > > We thank the reviewer for the quick response. Following the reviewer’s constructive comments, we ran the required experiments and further revised the paper by adding the new experimental results. Below, we respond to the reviewer comments by stating the new experiments and summarizing the corresponding results.
> > >
> > > 1. The reviewer suggested adding the $p$-values on the differences for the box-plots. Following this suggestion, we compared the fidelity/sensitivity difference between our method and the corresponding baseline approach. To be more specific, given two sets of fidelity/sensitivity scores ($s_{\text{our}}$ and $s_{\text{base}}$) obtained from our method and a baseline approach respectively, we first computed their difference, i.e., $diff = s_{\text{our}} - s_{\text{base}}$. Then, we conducted a statistical measure on the values of $diff$ by computing the mean, the standard error, and the $p$-value of the paired t-test. For the paired t-test, our null hypothesis is $H_{0}: \mathbb{E}[diff] \geq 0$. This indicates that, if the value of $p$ is larger than a threshold, we cannot reject this null hypothesis, and have to conclude that our method cannot outperform the corresponding baseline approach (i.e., achieving lower fidelity scores). In our newly revised submission, for the $diff$ we computed above, we present the mean, standard deviation, and p-value for every box-plot shown in Fig. 2(B), Fig. 3(B), Fig. 4(B)$\sim$(D), Fig. A9, Fig. A13(a), Fig. A13(b), Fig. A14, Fig. A15. We updated our writeup in Section A19 to reflect these changes. As we present in Table A4$\sim$Table A11, the overall experimental results align with those shown in the box plots, demonstrating the superiority of our method over the baselines. But, it should also be noted that we observed four cases in the SST experiment (see Table A5), where the p-value is larger than 0.5. This implies that, while our method outperforms existing baseline methods and alternative designs in general, for some cases, alternative designs (e.g., using constants/noises to replace decoys) may still have their advantages. We thank the reviewer for pointing out the necessity of adding this new experiment, which helps us identify this issue. As part of our future work, we will take a closer look at these cases and investigate the reason hidden behind this observation.

---

### Official Review · AnonReviewer1 · 2020-10-30
**A saliency map method based on "decoys"**

**Rating:** 6
**Confidence:** 2

**Review:**

Summary of work: a new saliency map method is presented, along with empirical work showing that it beats SoTA by some metrics.

Recommendation: I lean toward accepting this paper.

Reasoning:

Strengths: the paper represents an advance in state of the art, based on a variety of quantitative metrics. The authors have come up with an interesting new idea, and done a fair amount of work to back up their claims. It would be useful to have this paper in the literature.

Weak points: Interpretability methods ultimately stand or fall based on how useful they are to humans, not quantitative metrics. To my eye, the results for images and sentences aren't noticeably better than the alternatives. Furthermore, I found the writing fairly confusing--I needed to read the "decoy" definition several times.

Areas for improvement:
* I'd like to see a better description of the basic decoy method, maybe with some figures walking us through the construction. I would also like to see some motivation for what feels like a fairly complex technique, beyond Theorem 1.
* Theorem 1 is suggestive, but I'd like to see a critical analysis of how it applies in the real world. Do we have any idea how big the constant $C_1$ is, for instance? Furthermore, $F^c$ is only piecewise differentiable, and on very small pieces in practice. (Also, if it weren't for the softmax layer, the function would be piecewise linear--it's not clear to me what the Hessian tells us about anything that happens before that final layer.) Given this, it's not obvious that the Hessian at a given point captures significant information about robust inter-feature interactions.
* I find Figure 3 confusing (e.g., there are undefined abbreviations) and visually it's not clear that the decoy method is better. Maybe having one large diagram, with callout arrows and annotations, would help make it clear what the new method is adding.
* I did not check the math in the appendix (if I understand correctly, this is not necessary in a review, but I wanted to make that fact explicit.)

---

> ### Author Response · Authors · 2020-11-22
> **[Responses to AnonReviewer1]-Responded to the reviewer's questions about the motivation and technique; added illustration figures; Updated the visualization figures based on the reviewer's suggestion.**
>
> We would like to thank the reviewer for reviewing our paper. A point-by-point response is shown below.
>
> 1. The reviewer suggested using some figures to illustrate the construction of decoys. We made the changes to our submission by drawing two figures (Fig 1 A and B). Fig. 1(A) is a schematic workflow of the proposed method. Fig. 1(B) is an illustration of the swap operation -- the core idea in decoy construction.
>
> 2. The reviewer asked about some motivation for the method beyond the theorem. We divide the motivation into two aspects: the design of decoys and the range aggregation. First, for the design of decoys, because we noted that the existing perturbation potentially makes the image out-of-distribution and subsequently causes the classifier behavior to be ill-defined [1], we designed decoys to preserve the distribution by resembling the intermediate representation of the original image. Second, as we showed in the paper, with the range aggregation, our method enables good empirical performance. In addition, parallel work in statistics also demonstrates the effectiveness of using a range-based metric to control the false discovery rate in variable selection[2].
>
> 3. The reviewer asked how big the constant $C_1$ is in Theorem 1. As shown in Section A7, $C_1$ is not only dependent on the input but also the intrinsic behavior of the neural network (i.e. Lipschitz continuity [3]). We added descriptions in Section A7 and stated that we will study the tightness of this bound as part of our future work.
>
> 4. The reviewer questioned whether the Hessian could capture sufficient information about robust inter-feature interactions in neural networks. We believe that the Hessian can capture the inter-feature interaction information for the following reasons. First, in this paper, we consider the neural network with the softmax as the final layer. With this setup, the neural network is not precisely piecewise linear, and the Hessian is meaningful. Second, Eqn. (33) shows that the Hessian includes the neighborhood features that are jointly activated, indicating inter-feature interaction. Finally, existing work suggests that the Hessian is meaningful for deep neural networks with softmax[4].
>
> 5. The reviewer pointed out that some figures are confusing and visually unclear. We made two changes and updated all the experiment results. For the first change, we followed the second reviewer’s suggestion. We retain only the top 20% of pixels ranked by each saliency method for visualization, instead of plotting the saliency values for all pixels. Second, we visualize the difference of saliency methods with and without using our decoy method.
>
> [1]“Interpreting Black Box Models via Hypothesis Testing”.
> [2]“Neural Gaussian Mirror for Controlled Feature Selection in Neural Networks”.
> [3]“Intriguing properties of neural networks”.
> [4]“Understanding impacts of high-order loss approximations and features in deep learning interpretation”.

---

### Decision · Program_Chairs · 2021-01-07
**Final Decision**

**Decision:**

Reject

**Comment:**

This paper presents a novel approach to producing saliency maps for interpreting deep neural networks.  In general this paper seems quite close to borderline, although on the positive side, with some low confidence reviews.  The reviewers felt that the proposed approach could be useful to the community and they seemed to feel that the qualitative results in the experiments demonstrated convincing saliency maps.  There were some concerns, however, about the quantitative experiments as the reviewers (e.g. AnonReviewer2) found that while the mean results seemed better, it wasn't clear if they were statistically significant.  Naturally, the qualitative experiments are highly subjective and there was disagreement between reviewers whether the proposed approach did indeed produce better saliency maps than existing approaches such as smoothgrad.  One reviewer indicated that they found it difficult to follow the paper and to understand the decoy concept given the writing.  During discussion AnonReviewer2 updated their score (and very thorough review) by 2 points to indicate a weak preference toward accept.  None of the reviewers argued particularly strongly for acceptance and "championed" the paper.

The low confidence, slightly above borderline reviews seem to suggest that the reviewers thought the paper was above the bar but were reluctant to argue strongly for acceptance.  The method seemed like it could be useful to them but they weren't clearly convinced that it set a new state-of-the-art given the quantitative and qualitative empirical results.